# ER stress induced mitochondrial dysfunction drives Treg instability in coronary artery disease

Smriti Parashar [1✉], Mohammad Oliaeimotlagh[1,4,6], Payel Roy[1,5,6], Qingkang Lyu[1], Anusha Bellapu[1], Mikhail Fomin[1], Sunil Kumar[1], Yan Wang[1], Chantel C McSkimming[2,3], Coleen A McNamara[2,3] & Klaus Ley [1✉]

## Abstract

Under conditions of chronic unresolved inflammation characteristic of atherosclerosis, regulatory CD4+ T cells (Tregs) become unstable and convert to cytotoxic exTregs. The mechanism driving this conversion in humans is unclear. Here, we show unresolved endoplasmic reticulum (ER) stress as a key factor driving Treg instability. Human exTregs undergo ER stress and consequent mitochondrial dysfunction that remains unchecked due to defective mitophagy. Integrated stress response (ISR), a pathway that can trigger inflammatory signaling, is also upregulated in exTregs. exTregs are highly apoptotic and are more susceptible to stress-mediated cellular dysfunction due to their senescent state. In a phenotype reminiscent of exTregs, Tregs from coronary artery disease (CAD) patients show high ER stress and mitochondrial depolarization. This is further exacerbated in CD4+ T cells residing in atherosclerotic plaques. Pro-atherosclerotic stressors such as oxLDL and interferon-γ induce ER stress and mitochondrial dysfunction in Tregs in vitro. We conclude that the maladaptive inflammatory environment in atherosclerosis triggers ER stress and mitochondrial dysfunction, contributing to Treg instability in CAD.

**Keywords** Tregs; exTregs; ER Stress; Mitochondrial Dysfunction; Atherosclerosis
**Subject Categories** Cardiovascular System; Immunology

## Introduction

Regulatory T cells (Tregs) are a specialized kind of suppressive T cells characterized by expression of the lineage-defining transcription factor FOXP3 (Fontenot et al, 2003; Hori et al, 2003; Khattri et al, 2003; Sakaguchi et al, 1995; Takahashi et al, 1998). Congenital deficiency of Tregs leads to fatal systemic autoimmunity, and Treg dysfunction is linked to numerous inflammatory diseases (Dikiy and Rudensky,

2023). Chronic inflammation can cause Treg instability and drive them toward exTregs that exhibit effector T cell-like functions (Butcher et al, 2016; Saigusa et al, 2022; Wolf et al, 2020). Our recent work for the first time identified the phenotype of human exTregs as CD3 + CD4 + CD8- CD56 + CD16+ cells that share TCR CDR3 sequences with Tregs (Freuchet et al, 2023). Human exTregs are not suppressive and instead gain inflammatory and cytotoxic properties (Freuchet et al, 2023). The cellular mechanisms that drive the conversion of Tregs to exTregs in humans are not known.

High expression of cytotoxic and inflammatory proteins like perforin, granzyme B, tumor necrosis factor (TNF), and interferon gamma (IFNγ) in exTregs suggests that they are actively secreting cells (Freuchet et al, 2023). Secretory overload can be detrimental to cells (Hetz et al, 2020). The endoplasmic reticulum (ER) is a central hub for protein synthesis, folding, and post-translational modifications (Walter and Ron, 2011). During a high secretory state, the protein-folding machinery in the ER gets overwhelmed, leading to errors and subsequent accumulation of aberrant proteins in the ER lumen (Walter and Ron, 2011). In addition, physiological conditions such as nutrient excess and an inflammatory environment can also disrupt ER homeostasis (Bettigole and Glimcher, 2015). Failure to clear misfolded proteins can lead to ER stress (Walter and Ron, 2011). Prolonged unresolved ER stress can drive cells towards apoptosis (Malhotra and Kaufman, 2011).

The ER maintains close contact with mitochondria via specialized sites called ER-mitochondria contact sites (ERMCS) (Rowland and Voeltz, 2012). ERMCS are used to transfer calcium and lipids to mitochondria, thus regulating mitochondrial function and dynamics (Malhotra and Kaufman, 2011). Increased ER–mitochondrial coupling is observed during early phases of ER stress, causing enhanced calcium transfer and a resultant boost in mitochondrial bioenergetics (Bravo et al, 2011). However, prolonged influx of calcium into mitochondria can lead to the opening of mitochondrial permeability transition pores (PTP), causing mitochondrial dysfunction and release of a series of pro-apoptotic proteins into the cytosol (Bauer and Murphy, 2020; Baumgartner et al, 2009; Malhotra and Kaufman, 2011).

Mitochondrial dysfunction can affect Treg stability (Alissafi et al, 2020; Desdin-Mico et al, 2020; Weinberg et al, 2019).

[1]Immunology Center of Georgia, Augusta University, Augusta, GA, USA. [2]Carter Immunology Center, University of Virginia, Charlottesville, VA, USA. [3]Cardiovascular Research Center, Division of Cardiovascular Medicine/Department of Medicine, University of Virginia, Charlottesville, VA, USA. [4]Present address: Perelman School of Medicine, University of Pennsylvania, Philadelphia, PA, USA. [5]Present address: Department of Biochemistry, Indian Institute of Science, Bangalore, Karnataka, India. [6]These authors contributed equally: Mohammad Oliaeimotlagh, Payel Roy. ✉E-mail: sparashar@augusta.edu; kley@augusta.edu

Dysfunctional mitochondria are cleared by a selective autophagic process called mitophagy (Youle and Narendra, 2011). The inability to clear defective mitochondria can lead to oxidative stress-induced apoptosis, a type of programmed cell death which can be beneficial or harmful for immune cells depending on the context (Brokatzky et al, 2019; Vringer and Tait, 2023).

Treg instability is associated with various autoimmune and inflammatory pathologies (Dikiy and Rudensky, 2023). Coronary artery disease (CAD), a disease with a strong autoimmune component, is a leading cause of death worldwide (Kimura et al, 2018; Kimura et al, 2017; Vaduganathan et al, 2022; Wolf et al, 2020). CD4 + T cells, as central regulators of cellular and humoral immune responses, have a central role in CAD progression (Roy et al, 2022). They have recently been shown to be a major immune cell subset in atherosclerotic plaques (Depuydt et al, 2023; Fernandez et al, 2019). While inflammatory CD4 + Th1 cells are atherogenic, regulatory Treg cells are atheroprotective (Ait-Oufella et al, 2006; Klingenberg et al, 2013; Mor et al, 2006; Mor et al, 2007). Patients with CAD have high circulating levels of inflammatory cytokines like TNF and IFNγ which can trigger loss of protective Treg functions (Ait-Oufella et al, 2006; Butcher et al, 2016; Hansson and Libby, 2006; Khan et al, 2024). However, the intrinsic pathways driving Treg plasticity in CAD remain poorly understood.

In this study, we investigated the mechanisms steering the conversion of Tregs to cytotoxic exTregs in humans. We considered exTregs as the end-product of Treg instability and analyzed key cellular quality control pathways, which we found to be perturbed in these cells by combining immunological, cell biological, and bioinformatic approaches. We identified unresolved ER stress and loss of mitochondrial function as causal players driving Treg instability in humans. To understand the relevance of these findings in the context of chronic inflammatory diseases, we analyzed the Tregs from patients with CAD and CD4 + T cells in human atherosclerotic plaques. We found disturbed ER and mitochondrial homeostasis in these cells, a phenotype typical of human exTregs, explaining the loss of Treg function in inflammatory immune pathologies.

## Results and discussion

### ER stress drives instability of human Tregs

Tregs in patients with metabolic disorders like atherosclerosis are chronically exposed to nutritional and inflammatory triggers (Ajoolabady et al, 2024) that can disrupt ER homeostasis. As Tregs convert to inflammatory intermediates (Butcher et al, 2016) and exTregs (Freuchet et al, 2023), they gain a highly secretory phenotype, which, if overwhelming, is a known trigger for ER stress (Hetz et al, 2020). We hypothesized that ER stress might be a cause of Treg instability and, consequently, exTregs might display disrupted ER homeostasis.

To test this, we analyzed the bulk RNA transcriptomes from a published dataset of sorted human Tregs (CD3 + CD8- CD4 + CD25+ CD127lo) and exTregs (CD3 + CD8- CD4 + CD56 + CD16 + ) (Freuchet et al, 2023). Differential gene expression analysis showed significant upregulation of genes involved in protein secretion (RAB1A, RAB2A, RAB2B) and ER stress (HSPA5, DNAJC3, ATF6B, XBP1) in exTregs (Fig. 1A). Gene set enrichment analysis (GSEA) also showed enrichment of the hallmark signature for unfolded protein response (UPR) in these cells (Fig. 1B). XBP1 and ATF6B, key transcription factors involved in orchestrating response to ER stress, were transcriptionally upregulated in exTregs (Fig. EV1A). In addition, expression of the ER chaperone BiP (encoded by HSPA5), a major sensor of ER stress, was significantly higher at both transcript (Fig. EV1A) and protein level in exTregs (Fig. 1C, gating strategy in Fig. EV1B). Expression of IRE1 and PERK, critical mediators of UPR signaling, was also high in exTregs (Figs. 1D,E and EV1C,D). These findings suggest that exTregs are in a state of ER stress.

Another consequence of ER stress is ER expansion, which is the cell's attempt to accommodate the newly synthesized protein-folding machinery (Schuck et al, 2009). Consistent with this, exTregs showed enhanced staining for ER tracker, a dye that labels ER membranes, by flow cytometry (Fig. 1F). Prolonged ER stress can activate the expression of CHOP (transcription factor C/EBP homologous protein) and trigger apoptosis (Hetz et al, 2020). A sevenfold higher percentage of exTregs (~27%) expressed CHOP compared to Tregs (~4%) (Figs. 1G and EV1E), suggesting that ER stress remains unresolved in a significant proportion of these cells. To test whether protein misfolding triggered by high secretory load is causing ER stress in exTregs, we analyzed the accumulation of proteostat, a dye that fluoresces when it gets intercalated into aggregated proteins, by flow cytometry. Indeed, exTregs showed enhanced fluorescence of proteostat compared to Tregs (Fig. 1H). MG132, a proteasomal inhibitor, was used as a positive control (Figs. 1H and EV1F).

As exTregs are a product of Treg instability (Freuchet et al, 2023), we checked if direct induction of ER stress can drive this phenotype. We generated induced Tregs (iTregs) in vitro (Fig. EV1G), and treated them with tunicamycin, a known inducer of ER stress (Oslowski and Urano, 2011). Western blot analysis showed a gradual increase in expression of both BiP and CHOP in iTregs treated with tunicamycin (Fig. 1I). Loss of FOXP3 is a hallmark of Treg instability, and tunicamycin-treated iTregs showed a decrease in the frequency of FOXP3+ cells (Fig. 1J). A similar decrease was observed in the frequency of CD25 + FOXP3+ Tregs (Figs. 1K and EV1H) and CD25+CD127lo Tregs (Fig. EV1I) in human PBMCs treated with tunicamycin. Additionally, tunicamycin reduced FOXP3 expression in both natural Tregs (nTregs) and induced Tregs (iTregs) that were generated from the same donors (Fig. EV1J), suggesting that ER stress can drive FOXP3 loss in both subsets of Tregs. Whether this loss is tied to CNS2 methylation status in these subsets remains to be addressed.

Taken together, our findings suggest that persistent, unresolved ER stress can drive Treg instability in humans and potentially steer them toward exTregs. In mice, Treg-specific ablation of Hrd1, a key mediator of ER-associated degradation, causes the loss of the Treg's immunosuppressive properties under inflammatory conditions (Xu et al, 2019). Thus, we conclude that failure to mitigate ER stress can contribute to Treg plasticity in chronic inflammatory diseases.

### ER stress-induced mitochondrial dysfunction triggers Treg instability

ER maintains close contact with mitochondria via ER-mitochondria contact sites (Rowland and Voeltz, 2012). These

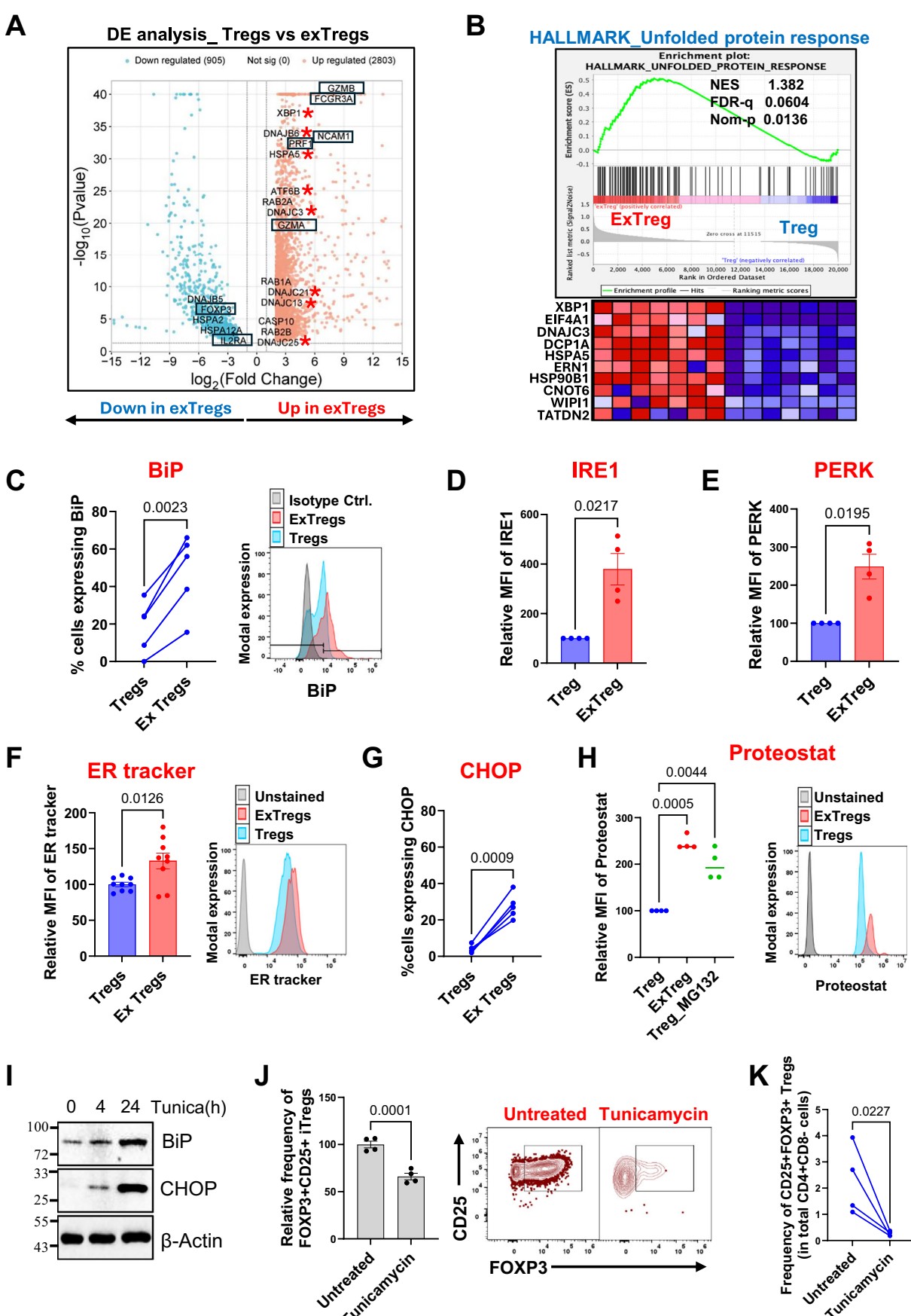

**Figure 1. Endoplasmic reticulum stress drives Treg instability.**

(A) Volcano plot representing significantly differentially expressed genes between human Tregs and exTregs. Left, down in exTregs (blue). Right, up in exTregs (red). Genes coding for endoplasmic reticulum (ER) and cytosolic chaperones are marked with asterisks. Canonical Treg and exTreg genes are shown in boxes. y and x axes are capped at 40 ($P = 10^{-40}$) and $+/-$ 15 (log2FC), respectively. Statistical analyses were performed using a two-tailed Wald test with Benjamini–Hochberg P value adjustment. (n = 7 human subjects). (B) GSEA plot showing enrichment of the Hallmark gene signature for unfolded protein response (M5922 in mSigDB) within exTreg and Treg (n = 7) transcriptomes. Normalized enrichment score (NES), FDR q and nominal P values are indicated. A heatmap for the top ten enriched genes in exTregs vs Tregs is shown. Color scale in the heatmap is based on GSEA row minimum (blue) to row maximum (red). (C) Human PBMCs were stained for intracellular BiP (*HSPA5*) and analyzed by flow cytometry. Representative histograms (right) show the mean fluorescence intensity (MFI) of BiP in Tregs (blue), exTregs (red) and isotype control (gray). The y axis was normalized to mode. Left: Bar graph showing frequency of Tregs and exTregs expressing BiP. n = 5. Each dot represents a biological replicate from an independent human donor. (D, E) Human PBMCs were stained for intracellular IRE1 (D) and PERK (E) and analyzed by flow cytometry. MFI of these proteins in Tregs from each donor was normalized to 100 and relative MFI in exTregs from the respective donor was plotted against it. n = 4. Each dot represents a biological replicate from an independent human donor. (F) Human PBMCs were stained with ER tracker red and analyzed by flow cytometry. The average MFI of ER tracker in Tregs from all donors in an experiment was normalized to 100, and the relative MFI of ER tracker in Tregs and exTregs from each donor was plotted against it. Each dot represents a biological replicate from an independent human donor. n = 9. Representative histograms are shown on the right. The y axis was normalized to the mode. (G) Frequency of Tregs and exTregs staining intracellularly for CHOP by flow cytometry. n = 5. Each dot represents a biological replicate from an independent human donor. (H) Human PBMCs were intracellularly stained with Proteostat and analyzed by flow cytometry. MFI of proteostat in Tregs from each donor was normalized to 100 and relative MFI of Proteostat in exTregs from the respective donor was plotted against it. PBMCs treated with MG132 were used as positive control. The horizontal bars represent the median. n = 4. Each dot represents a biological replicate from an independent human donor. (I) In vitro induced human Tregs (iTregs) were treated with tunicamycin, and the expression level of BiP and CHOP was analyzed by western blot. Molecular weights (kDa) are indicated on the left of the blots. β Actin was used as a loading control (J) iTregs were treated with tunicamycin for 24 h, and the frequency of CD25 + FOXP3+ cells was compared to untreated control. n = 4. Representative flow cytometry plots are shown on the right. (K) Frequency of CD25 + FOXP3+ Tregs in human PBMCs treated with tunicamycin for 72 h. Untreated PBMCs were used as a control. n = 4. Each dot represents a biological replicate from independent human donors. Statistical comparisons were done using paired two-tailed T test (C–G, J, K) and two-way ANOVA with Bonferroni's multiple comparisons in (H). The results are represented as mean ± SEM in the bar graphs. Numerical P values are listed at the top of each graph. Source data are available online for this figure.

sites are critical for lipid exchange as well as calcium transfer from ER to mitochondria (Rowland and Voeltz, 2012). Early ER stress leads to enhanced coupling of the ER with mitochondria to boost the generation of energetic substrates essential for cellular adaptive response to this stress (Bravo et al, 2011; Gottschalk et al, 2022). However, prolonged coupling due to unresolved ER stress can cause mitochondrial depolarization, which leads to apoptosis (Bravo et al, 2011).

To determine the consequences of ER stress on mitochondrial function in Tregs, we treated PBMCs with tunicamycin and analyzed the retention of tetramethyl rhodamine methyl ester (TMRM), a dye that accumulates in active mitochondria with an intact membrane potential (Wculek et al, 2023). We found that prolonged (48 h) but not short (24 h) tunicamycin treatment caused a significant increase in the frequency of Tregs with lower mitochondrial membrane potential (Figs. 2A and EV2A; Appendix Fig. S1). In addition, tunicamycin treatment also led to accumulation of Tregs with enhanced mitochondrial reactive oxygen species (mitoROS) (Figs. 2B and EV2B). Interestingly, almost all Tregs with high mitoROS had low TMRM retention (Fig. EV2C), suggesting mitoROS as triggers for mitochondrial depolarization in these cells. Consistent with this, a previous study has shown that mitoROS accumulation can drive loss of mitochondrial membrane potential in cardiac myocytes (Zorov et al, 2000). Another hallmark of mitochondrial dysfunction is the accumulation of PINK1 (PTEN-induced kinase 1) on damaged mitochondria. This triggers mitophagy, a protective response that removes dysfunctional mitochondria by targeting them to the lysosomes (Youle and Narendra, 2011). However, we found a marked reduction in overall protein expression of PINK1 in tunicamycin-treated Tregs (Fig. EV2D). Indeed, ATF3-mediated transcriptional repression of PINK1 during conditions of ER stress has been shown to disrupt mitochondrial homeostasis in alveolar cells of patients with idiopathic pulmonary fibrosis (Bueno et al, 2018; Bueno et al, 2015). Taken together, our findings suggest that persistent ER stress can perturb mitochondrial homeostasis in Tregs.

To further explore the consequences of ER stress on Tregs, we assessed mitochondrial mass in tunicamycin-treated cells by tracking the accumulation of mitotracker green, a dye that labels all mitochondria irrespective of the membrane potential. We observed a subtle but significant decrease in mitochondrial mass in Tregs after tunicamycin treatment (Fig. EV2E). Tunicamycin treatment also caused an increase in the percent of Tregs with elongated mitochondrial morphology as observed by live-cell confocal imaging (Figs. 2C and EV2F). Mitochondrial elongation has been proposed as a structural adaptation to facilitate mitochondrial respiration under cellular stress (Eisner et al, 2018; Lebeau et al, 2018). Consistent with this, we found that tunicamycin-treated Tregs still relied on mitochondria as their major source of energy production (Fig. EV2G) as determined by SCENITH (Arguello et al, 2020). We conclude that while under our experimental conditions, tunicamycin-mediated ER stress has begun to induce mitochondrial dysfunction in Tregs (~15% cells are TMRM- and MitoROS + ), the effect is not absolute yet, and the cells still have the capability to adapt to this stress. However, we speculate that this adaptability may not sustain during chronic inflammatory pathologies, wherein Tregs are exposed to ER stress much longer than our experimental conditions.

As exTregs are in a state of unresolved ER stress (concluded from Fig. 1), we hypothesized that this may cause mitochondrial dysfunction in these cells. Indeed, we found a marked reduction in retention of TMRM in exTregs (Figs. 2D and EV2H). Mitochondrial dysfunction can result in increased mitochondrial reactive oxygen species (mitoROS) production (Sena and Chandel, 2012). Consistently, exTregs showed a 3.5-fold higher mitoROS staining compared to Tregs (Fig. 2E) and an overall enrichment of gene signature for reactive oxygen species (Fig. EV2I).

To determine if mitophagy can clear dysfunctional mitochondria in exTregs, we analyzed the expression of mitophagy-related genes in these cells. There were significantly reduced transcripts of *PINK1*, *PRKN* (Parkin), *SQSTM1*(p62), *OPTN* (Optineurin), and *MAPLC3B1*

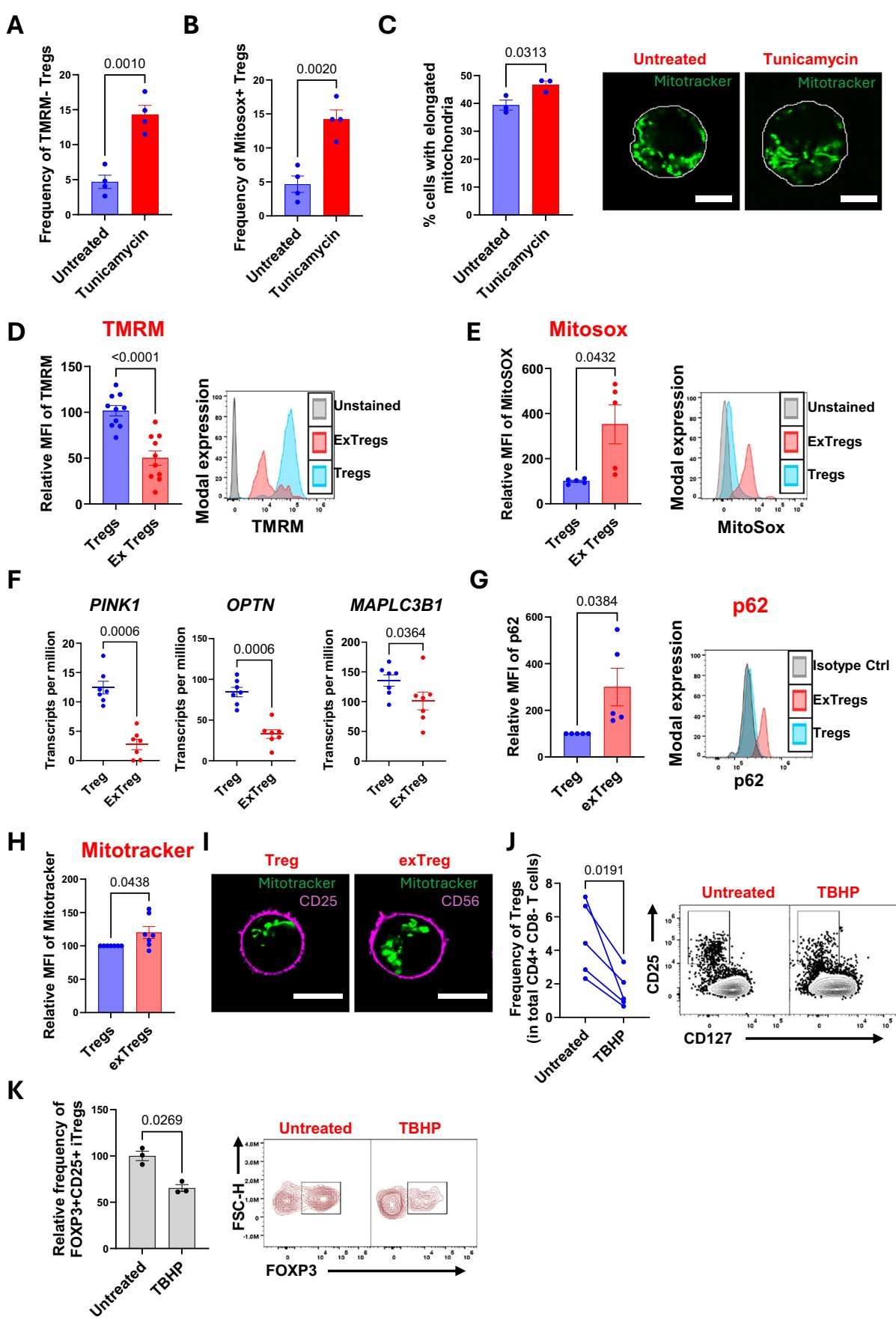

**Figure 2. Endoplasmic reticulum stress drives mitochondrial dysfunction in Tregs.**

(A, B) Human PBMCs were treated with tunicamycin for 48 h and stained for TMRM and Mitosox. The graph compares the frequency of Tregs with low TMRM retention (A) and high Mitosox staining (B) in untreated vs tunicamycin-treated samples. Each dot represents a biological replicate from an independent human donor. $n = 4$. (C) Tregs were cultured in vitro and treated with Tunicamycin for 48 h. Mitochondria were labeled with Mitotracker green and visualized by live-cell confocal imaging. Untreated Tregs were used as a control. The bar graph shows % of Tregs that showed tubular/elongated morphology. Each dot represents a biological replicate from an independent human donor. $n = 3$. Representative images are shown on the right. Scale bar, 5 µm. In all, 15–30 cells were analyzed from each donor for each condition. (D) Human PBMCs were stained with TMRM and analyzed by flow cytometry. The average MFI of TMRM in Tregs from all donors in an experiment was normalized to 100, and the relative MFI of TMRM in Tregs and exTregs from each donor was plotted against it. Each dot represents a biological replicate from an independent human donor. $n = 10$. Right, representative histograms showing the fluorescence intensity of TMRM in Tregs (blue), exTregs (red), and unstained cells (gray). The y axis was normalized to the mode. (E) Human PBMCs were stained for MitoSox and analyzed by flow cytometry. MFI in Tregs and exTregs was plotted as explained in (D). $n = 5$. Right, Representative histograms showing the fluorescence intensity of MitoSox in Tregs (blue), exTregs (red), and unstained cells (gray). The y axis was normalized to the mode. (F) Normalized expression levels (transcripts per million) of mitophagy genes *PINK1*, *OPTN*, and *MAPLC3B1* in human bulk transcriptomes from sorted human Tregs and exTregs. Horizontal bars represent the median. $n = 7$. (G) Human PBMCs were intracellularly stained for p62. MFI of p62 in Tregs from each donor was normalized to 100, and the relative MFI of p62 in exTregs from the respective donor was plotted against it. $n = 4$. Each dot represents a biological replicate from an independent human donor. (H) Human PBMC's were stained with Mitotracker and analyzed by flow cytometry. MFI of mitotracker in Tregs from each donor was normalized to 100, and relative MFI of mitotracker in exTregs from the respective donor was plotted against it. Each dot represents a biological replicate from an independent human donor. $n = 7$. (I) Mitochondrial morphology in Tregs and exTregs was compared by staining with Mitotracker Green. Representative confocal images are shown. Scale bar, 5 µm. (J) Frequency of CD25+CD127lo Tregs in human PBMCs treated with TBHP for 24 h. Untreated PBMCs were used as a control. $n = 5$. Each dot represents a biological replicate from an independent human donor. Representative flow cytometry plots are shown on the right. (K) iTregs were treated with TBHP for 4 h, and their frequency was compared to untreated control. $n = 3$. Representative flow cytometry plots are shown on the right. Statistical comparisons in (A–E, G, H, J, K) were done using a paired two-tailed T test and in (F) using two-tailed Mann–Whitney U test. Results are represented as mean ± SEM. Numerical P values are listed at the top of each bar graph. Source data are available online for this figure.

(Figs. 2F and EV2J) in exTregs. In addition, while being reduced at the transcript level, we found enhanced accumulation of p62 protein in exTregs, suggesting impaired mitophagy (Fig. 2G). Consequently, we observed enhanced overall mitochondrial mass in exTregs as determined by significantly higher mitotracker staining (Figs. 2H and EV2K) as well as higher frequency of exTregs that showed accumulation of dysfunctional mitochondria (Fig. EV2L). In addition, mitochondrial morphology in exTregs appeared to be swollen as revealed by live-cell confocal imaging, further supporting mitochondrial dysfunction in these cells (Fig. 2I).

Functionally, forcing mitochondrial depolarization by tert-butyl hydroperoxide (TBHP) reduced the frequency of Tregs (Fig. 2J) and induced an exTreg-like inflammatory phenotype as revealed by increased intracellular staining for IFNγ and TNF (Fig. EV2M). Treatment with TBHP alone was sufficient to downregulate FOXP3 expression in pure populations of iTregs, highlighting a causal association between disturbed mitochondrial membrane polarity and Treg instability (Fig. 2K).

Taken together, our findings suggest that persistent ER stress can drive mitochondrial dysfunction in Tregs and contribute to Treg instability. Clinically, enhanced mitochondrial oxidative damage is observed in Tregs of individuals with rheumatoid arthritis and systemic lupus erythematosus (Alissafi et al, 2020). Our findings identify ER stress to be a key trigger for mitochondrial dysfunction in Tregs that remains unchecked because of defective mitophagy.

## Human exTregs are apoptotic and terminally differentiated

Cumulative stress stimuli from ER and mitochondria can trigger the integrated stress response (ISR), an evolutionarily conserved pathway critical for maintaining cellular homeostasis (Costa-Mattioli and Walter, 2020). The ISR restores balance by reprogramming gene expression to facilitate the mitigation of proteostatic and oxidative stress (Costa-Mattioli and Walter, 2020).

GSEA analysis revealed enrichment of the ISR gene signature in exTregs (Figs. 3A and EV3A), indicating the activation of this protective cellular response pathway. Consistently, exTregs showed enrichment for the IFNγ response gene signature (Figs. 3B and EV3B), a previously reported consequence of ISR (Costa-Mattioli and Walter, 2020; Deng et al, 2004). To further confirm activation of ISR in exTregs, we checked the accumulation of ATF4, a transcription factor involved in regulating UPR$^{ER}$, mitochondrial stress response, as well as integrated stress response (Hetz et al, 2020; Pakos-Zebrucka et al, 2016; Quiros et al, 2017). Flow cytometry analysis revealed enhanced expression of ATF4 in exTregs (Fig. 3C). ATF4 can induce metabolic reprogramming by regulating the expression of PCK2, an enzyme that can convert TCA intermediates to glycolytic intermediates (Quiros et al, 2017). Indeed, metabolic profiling by SCENITH revealed significantly reduced mitochondrial dependence in these cells (Fig. 3D). Expression of ATF5, a key downstream effector of ISR during mitochondrial stress (Fiorese et al, 2016; Nargund et al, 2012) was also upregulated in exTregs (Figs. 3E and EV3C). Taken together, these findings suggest that ER stress and mitochondrial dysfunction have triggered ISR in exTregs that rewires them toward an inflammatory and glycolytic phenotype.

While initially a protective response, prolonged ISR can trigger apoptosis (Pakos-Zebrucka et al, 2016). Consistent with this, the gene signature for apoptosis was enriched in exTregs (Figs. 3F and EV3D), suggesting that ISR fails to resolve stress in these cells. Consistently, ten times more (~50%) exTregs were in early stages of apoptosis as revealed by Annexin V staining compared to only ~5% of Tregs (Fig. 3G). DNA fragmentation is another hallmark of apoptosis that can be detected by Terminal Deoxynucleotide Transferase dUTP Nick End Labeling (TUNEL). Compared to Tregs (~2%), a significantly higher proportion (~30%) of exTregs were TUNEL+ (Fig. 3H). To confirm that dysregulated ER and mitochondrial homeostasis are causal factors driving exTreg apoptosis, we compared the phenotypes of apoptotic (Annexin V+) and non-apoptotic (Annexin V−) exTregs. Apoptotic exTregs had higher ER stress (Figs. 3I and EV3E) and lower mitochondrial membrane potential (Figs. 3J and EV3F) compared to non-apoptotic exTregs.

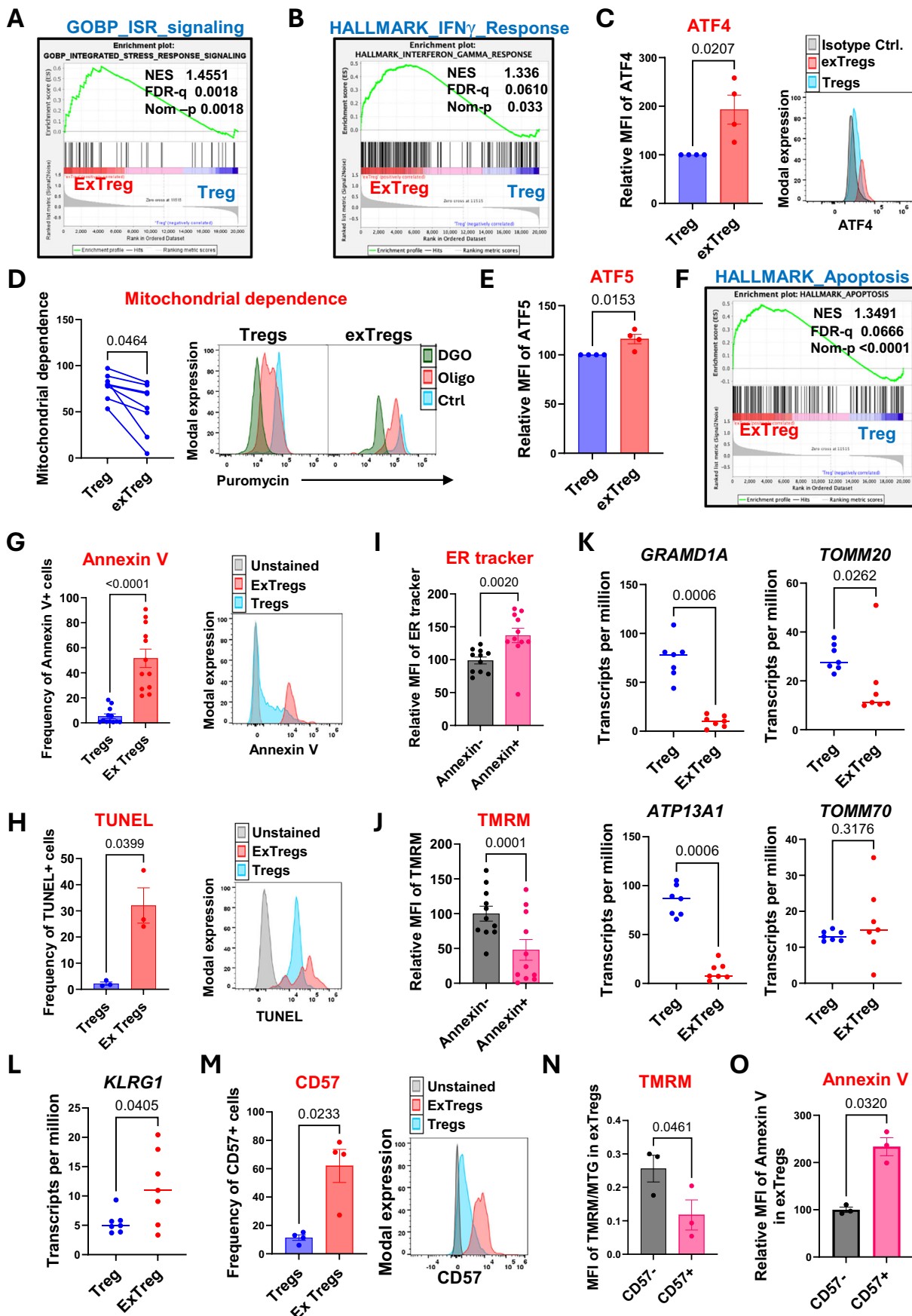

**Figure 3.  Human exTregs are apoptotic and terminally differentiated.**

(A, B) GSEA plots showing enrichment of gene signatures for ISR (GO:0140467) in (A) and IFNγ response (M5913 in mSigDB) in (B) within paired exTreg and Treg ($n = 7$ human subjects) transcriptomes. Normalized enrichment score (NES), FDR $q$, and nominal $P$ values are indicated. (C) Human PBMCs were intracellularly stained for ATF4 and analyzed by flow cytometry. MFI of ATF4 in Tregs from each donor was normalized to 100, and the relative MFI of ATF4 in exTregs from the respective donor was plotted against it. $n = 4$. Each dot represents a biological replicate from an independent human donor. Representative histograms are shown on the right. (D) Human PBMCs were analyzed for incorporation of puromycin in the absence or presence of various inhibitors by flow cytometry. The mitochondrial dependence of Tregs and exTregs was calculated as explained in the methods and is plotted in the bar graph. Histograms on the right show fluorescence intensity of puromycin in Tregs and exTregs when treated with vehicle control (blue), Oligomycin (red), or 2-deoxy-D-glucose plus oligomycin (green). Each dot represents a biological replicate from an independent human donor. $n = 8$. (E) Human PBMCs were intracellularly stained for ATF5 and analyzed by flow cytometry. MFI of ATF5 in Tregs from each donor was normalized to 100, and the relative MFI of ATF5 in exTregs from the respective donor was plotted against it. $n = 4$. Each dot represents a biological replicate from an independent human donor. (F) GSEA plots showing enrichment of gene signature for apoptosis (M5902 in mSigDB) within paired exTreg and Treg ($n = 7$) transcriptomes. Normalized enrichment score (NES), FDR $q$, and nominal $P$ values are indicated. (G, H) Human PBMCs were stained for Annexin V (G) or TUNEL (H) and analyzed by flow cytometry. Bar graphs show the frequency of Tregs and exTregs staining positive for Annexin V, $n = 12$ (G), or TUNEL, $n = 3$ (H). Each dot represents a biological replicate from an independent human donor. Histograms on the right of the bar graph show the fluorescence intensity of Annexin V (G) or TUNEL (H) for Tregs (blue), exTregs (red), and unstained cells (gray). The $y$ axis was normalized to the mode. (I, J) Relative MFI of ER tracker (I) or TMRM (J) in annexin V-positive and annexin V-negative exTregs. Each dot represents a biological replicate from an independent human donor. $n = 11$. (K) Normalized expression levels (transcripts per million) of genes involved in mediating ER to mitochondria protein transfer in human bulk transcriptomes from sorted human Tregs and exTregs. Horizontal bars represent the median. $n = 7$. (L) Normalized expression levels (transcripts per million) of *KLRG1* in human bulk transcriptomes from sorted human Tregs and exTregs. $n = 7$. (M) Human PBMCs were stained for CD57 and analyzed by flow cytometry. Frequency of Tregs and exTregs staining for CD57 is plotted. $n = 4$. Histograms on the right show fluorescence intensity of CD57 in Tregs (blue), exTregs (red) and unstained cells (gray). The $y$ axis was normalized to the mode. (N, O) Ratio of MFI of TMRM/MTG (N) and MFI of Annexin V (O) in CD57-exTregs vs CD57+ exTregs. $n = 3$. Each dot represents a biological replicate from an independent human donor. Statistical comparisons were done using the paired two-tailed $T$ test (C–E, G–J, M–O) and using two-tailed Mann–Whitney $U$ test (K, L). Results are represented as mean ± SEM. Numerical $P$ values are listed at the top of each bar graph. Source data are available online for this figure.

To check if transport disruptions at ER-mitochondria contact sites are contributing to the observed apoptotic phenotype in exTregs, we analyzed the expression of ER-mitochondria contact site proteins and of proteins involved in the ER-SURF pathway. The ER-SURF pathway is a major mechanism by which mitochondrial precursor proteins are transported from the ER to mitochondria, and defects in it can lead to mitochondrial dysfunction (Hansen et al, 2018; Koch et al, 2024). GSEA analysis revealed enrichment of the gene signature of ER-mitochondria contact sites in exTregs (Fig. EV3G). This is consistent with previous reports that ER stress can induce enhanced coupling of ER and mitochondria (Bravo et al, 2011; Gottschalk et al, 2022). However, transcripts of proteins involved in mediating transfer of mitochondrial precursor proteins from ER to mitochondria such as GRAMD1A (mammalian homolog of Lam6) (Koch et al, 2024) and TOMM20 (Lalier et al, 2021) was significantly downregulated in exTregs when normalized to total transcripts per million (Fig. 3K). No change in expression of TOMM70 was observed (Fig. 3K). Expression of ATP13A1, the P5A-ATPase required for translocation of ER-stranded mitochondrial precursors to mitochondria in humans (McKenna et al, 2022) was also significantly reduced in exTregs (Fig. 3K). Taken in context with the existing literature, our findings suggest that transport disruptions at ER-mitochondria junctions might contribute to the pro-apoptotic response in exTregs.

Cellular senescence, a stress response program implicated in aging and immunity, can aid apoptosis as a definitive cell-cycle exit mechanism (Reimann et al, 2024). It can be triggered by both replicative and physiological stress (Ben-Porath and Weinberg, 2004). GSEA analysis revealed enrichment of gene signature for cellular senescence in exTregs (Fig. EV3H). Compared to Tregs, exTregs had higher expression of killer cell lectin-like receptor subfamily G member 1 (KLRG1) (Fig. 3L) and CD57 (Fig. 3M), known senescence markers in T cells (Brenchley et al, 2003; Voehringer et al, 2002).

Cell division can help resolve proteotoxic stress by clearing misfolded protein aggregates (Du et al, 2025; Vaubourgeix et al, 2015; Zhou et al, 2014). As senescent cells typically lose the ability to divide (Reimann et al, 2024) and we also previously found exTregs to be non-proliferative (Freuchet et al, 2023), we

hypothesized that the senescent state of exTregs further worsens their inability to resolve stress. Indeed, CD57+ exTregs had lower TMRM retention (Figs. 3N and EV3I) and higher Annexin V labeling (Figs. 3O and EV3J) compared to CD57− exTregs.

Our finding that ER and mitochondrial dysfunction can trigger ISR in exTregs can explain their acquisition of an inflammatory phenotype. Previous reports have shown that ISR can activate the proinflammatory transcription factor NF-κB, driving transcription of a large set of proinflammatory genes (Costa-Mattioli and Walter, 2020; Deng et al, 2004). In addition, ISR can mediate a shift from OXPHOS to glycolysis by triggering expression of enzymes required for converting TCA intermediates into glycolytic substrates (Cohen et al, 2015; Han et al, 2013; Linares et al, 2017). Indeed, a metabolic shift from OXPHOS to glycolysis is a hallmark of unstable Tregs (Weinberg et al, 2019).

## Tregs from CAD patients show ER stress and mitochondrial dysfunction

To test if the mechanism we found for Treg destabilization is relevant to inflammatory immune pathologies such as CAD, we checked the effect of oxLDL (oxidized low-density lipoprotein) and IFN-γ, known physiological stressors relevant to atherosclerosis (Hansson and Libby, 2006; Roy et al, 2022), on Tregs. Prolonged treatment of human PBMCs with oxLDL or IFN-γ in vitro caused a subtle reduction in Treg frequency compared to untreated controls (Fig. EV4A) and enhanced apoptosis (Fig. EV4B). Both these stressors induced ER stress in iTregs as indicated by enhanced expression of BiP and spliced Xbp1 in the treated cells (Fig. 4A). However, the effect was most remarkable in the cells treated with a combination of oxLDL and IFN-γ (Fig. 4A), a condition that better mimics CAD-like environment. Consequently, enhanced expression of ATF4 (Fig. 4A) and a significant increase in the frequency of Tregs with low TMRM retention was observed under conditions of dual exposure to OxLDL and IFN-γ (Figs. 4B and EV4C). However, the extent of response to these stressors varied between donors, which we hypothesize may be driven by differential expression of receptors to these stressors in

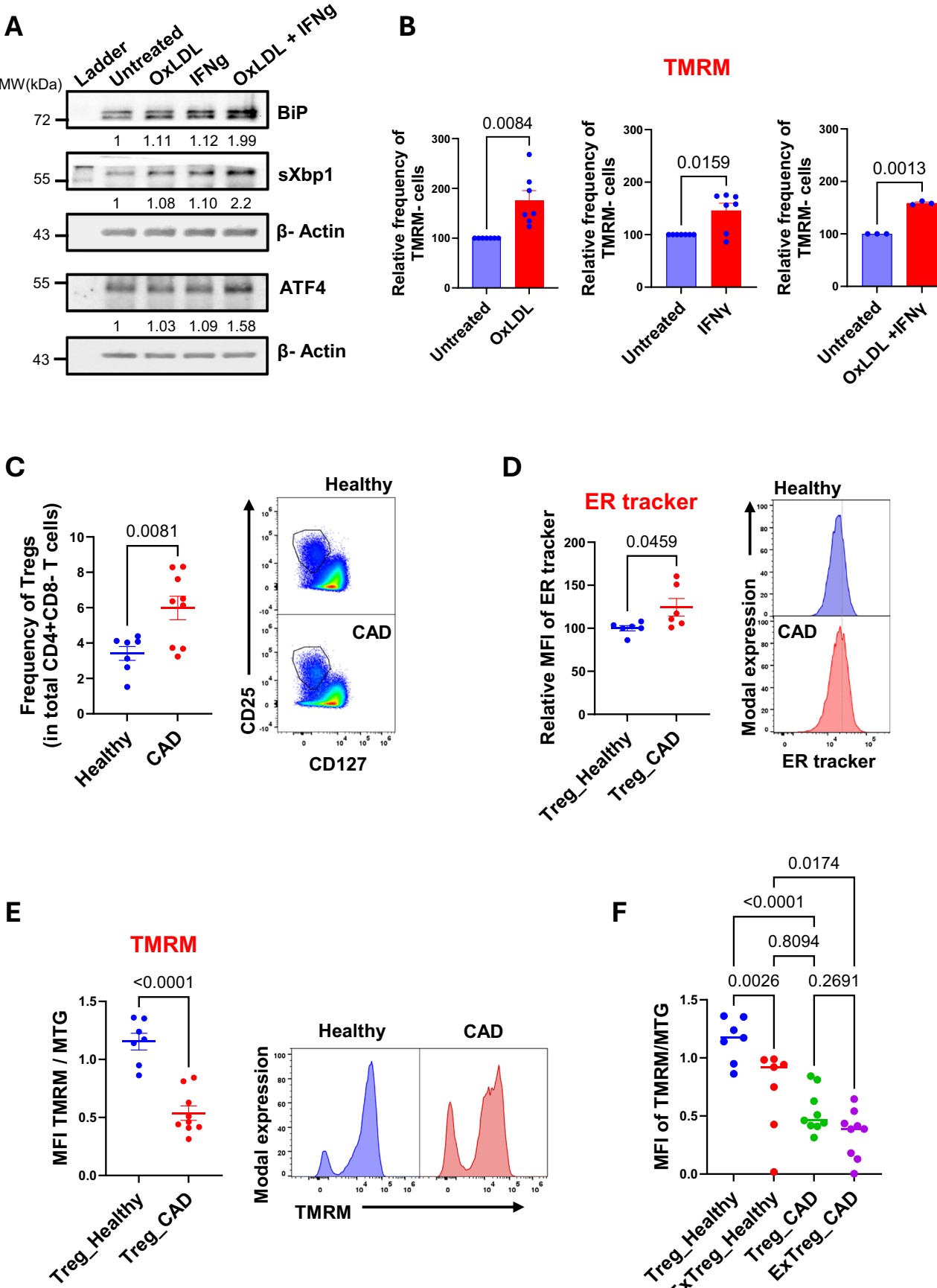

◄ **Figure 4. Tregs in CAD patients have ER stress and mitochondrial dysfunction.**

(A) In vitro-generated iTregs were treated with OxLDL, IFN γ or OxLDL + IFN γ and the protein expression of indicated ER stress markers was analyzed by western blotting. Molecular weights (kDa) are indicated on the left of the blots. β-actin was used as a loading control. Normalized levels of each protein to β-actin are indicated at the bottom of each blot. (B) Human PBMCs were treated with oxLDL, IFN γ, or oxLDL + IFN γ, and the relative frequency of Tregs with low TMRM retention (TMRM−) was plotted against untreated controls (normalized to 100). $n = 7$ for OxLDL and IFNγ, $n = 3$ for oxLDL + IFNγ. Each dot represents a biological replicate from an independent human donor. (C) Frequency of CD25 + CD127- Tregs in healthy and CAD subjects. $n = 7$ for healthy and $n = 9$ for patients. Each dot represents a biological replicate from an independent donor. Representative flow plots are shown on the right. (D) Bar graph comparing MFI of ER tracker in Tregs from healthy and CAD patients. $n = 6$. The average MFI of ER tracker in healthy Tregs from each experiment was normalized to 100, and the relative MFI of ER tracker in healthy and patient Tregs from each donor was plotted against it. Each dot represents a biological replicate from an independent donor. Representative histograms for healthy (blue) and CAD (red) are shown on the right. The dotted gray line is marked to show a distinct peak of cells showing high staining for ER tracker in CAD patients. (E) MFI of TMRM/MTG in Tregs from healthy and CAD patients. $n = 7$ for healthy and $n = 9$ for patients. Each dot represents a biological replicate from an independent donor. Representative histograms for healthy (blue) and CAD (red) are shown on the right. (F) Plot showing comparison of MFI of TMRM/MTG in Tregs and exTregs of healthy and CAD patients. $n = 7$ for healthy and $n = 9$ for patients. Statistical comparisons were done using paired two-tailed $T$ test in (B), unpaired two-tailed $T$ test in (C–E), and two-way ANOVA with Bonferroni's multiple comparison test in (F). Results are represented as mean ± SEM in (B). Horizontal bars represent the median in (C–F). Numerical $P$ values are listed at the top of each bar graph. Source data are available online for this figure.

different donors. These findings suggest that chronic exposure to physiological stressors implicated in CAD can trigger ER stress and mitochondrial dysfunction in Tregs.

To test this hypothesis, we analyzed ER stress and mitochondrial dysfunction in Tregs from the peripheral blood of CAD subjects (details in Table 2). Tregs from healthy donors were used as controls. First, we compared the frequency of circulating Tregs in healthy vs CAD patients and found significantly increased frequency of Tregs in patients (Fig. 4C). This suggests that it is not the loss of Tregs per se that drives the rogue immune response in CAD patients. Instead, compared to healthy controls, Tregs from CAD patients showed ER stress as revealed by higher ER tracker staining (Fig. 4D) and loss of mitochondrial membrane potential (Fig. 4E), a phenotype that we have now defined as typical of exTregs. Loss of TMRM retention in Tregs of CAD patients was so severe that it was not significantly different from their exTregs (Fig. 4F). This suggests that in CAD patients, some of the changes characteristic of exTregs have already begun to occur in Tregs. Indeed, studies have shown that Tregs from CAD patients are less suppressive and more inflammatory (Hasib et al, 2016), a phenotype that we have previously described as a hallmark of exTregs (Freuchet et al, 2023). Taken together, our findings suggest that chronic exposure to dietary and inflammatory stressors in CAD can trigger ER stress and mitochondrial dysfunction in Tregs that render them with an exTreg-like phenotype.

A catastrophic consequence of severe CAD is the rupture of atherosclerotic plaque triggered by highly inflamed microenvironment, causing thrombosis and organ damage (Hansson and Libby, 2006; Roy et al, 2022). We compared the single-cell transcriptomes (GSE196943) of matched blood and coronary artery plaque samples from patients showing advanced stages of plaque accumulation (Chowdhury et al, 2022). CD4 + T cells in plaques of these patients showed enriched expression of genes related to ER-stress and ISR, such as *HSPA5* (encoding BiP), *XBP1*, *ERN* (encoding IRE1), *DDIT3* (encoding CHOP), *EIF4A1*, and apoptosis (*CASP3*, *DDIT4*) (Fig. EV4D). We also found an overall increase in ER stress in circulating CD4 + T cells of CAD patients compared to healthy controls (Fig. EV4E). These findings suggest that perturbation of ER homeostasis may be a key driver of CD4 + T-cell dysfunction in CAD, a phenotype that is further exacerbated in atherosclerotic plaques.

In conclusion, we show the role of ER-stress-mediated mitochondrial dysfunction in human Treg to exTreg conversion. Human exTregs (CD3 + CD8-CD4 + CD56 + CD16 +) have enhanced ER stress, mitochondrial dysfunction, and enrichment of ISR signaling. We also show

that the acquisition of senescence markers like CD57 and KLRG1 by exTregs makes them more susceptible to stress. All these triggers eventually render exTregs apoptotic. This is the first study providing mechanistic insight into the transition of human Tregs to exTregs. We suspect that the low frequency of exTregs in human blood may be explained by their highly apoptotic phenotype. However, this limits their functional analysis to flow cytometric approaches, a technique that works well even with a low number of cells. Our study for the first time also implicates ER stress-induced mitochondrial dysfunction as a key driver of Treg dysfunction in CAD, a finding that may have important implications in understanding Treg instability in other autoimmune pathologies.

Implantation of Tregs in patients with autoimmune diseases or those undergoing transplants is being studied as an alternative approach to anti-inflammatory drugs (Bernaldo-de-Quiros et al, 2023; Bluestone et al, 2023). However, the destabilization of these implanted Tregs in the inflammatory environment remains a concern. Pharmacological or genetic interventions promoting the resolution of ER stress or ameliorating mitochondrial function in Tregs may have positive clinical implications.

# Methods

**Reagents and tools table**

| Reagent/resource | Reference or source | Identifier or catalog number |
|---|---|---|
| **Experimental models** | | |
| Healthy human PBMC's | Clinical core, La Jolla Institute for Immunology (LJI), La Jolla, California, USA | |
| PBMCs from CAD patients | University of Virginia Health System, Charlottesville, Virginia, USA | Coronary Assessment in Virginia (CAVA) cohort |
| **Recombinant DNA** | | |
| Not applicable | | |
| **Antibodies** | | |
| Alexa Fluor® 700 anti-human CD8a, Clone RPA-T8 | Biolegend | Cat.#301028, 1:100 |
| Pacific Blue™ anti-human CD4, Clone RPA-T4 | Biolegend | Cat. #300524, 1:100 |

| Reagent/resource | Reference or source | Identifier or catalog number |
|---|---|---|
| PE/Cyanine7 anti-human CD25, Clone M-A251, | Biolegend | Cat. #356107, 1:100 |
| CD127 Monoclonal Antibody (eBioRDR5), APC-eFluor™ 780, eBioscience™, | Thermo Fisher | Cat. #47-1278-42, 1:75 |
| PerCP/Cyanine5.5 anti-human CD3, Clone UCHT1 | Biolegend | Cat.#300430, 1:125 |
| Brilliant Violet 570™ anti-human CD45RO, Clone UCHL1 | Biolegend | Cat. #304226, 1:100 |
| Brilliant Violet 711™ anti-human CD45RO, Clone UCHL1 | Biolegend | Cat. #304235, 1:100 |
| Brilliant Violet 650™ anti-human CD16, Clone 3G8, | Biolegend | Cat.#302042, 1:75 |
| Brilliant Violet 785™ anti-human CD56 (NCAM), Clone 5.1 H11 | Biolegend | Cat. #362549, 1:75 |
| CD127 Monoclonal Antibody (eBioRDR5), Alexa Fluor™ 532, eBioscience™ | ThermoFisher | Cat. #58-1278-42, 1:75 |
| APC/Cyanine7 anti-human CD19, Clone H1B19 | Biolegend | Cat. #302217, 1:100 |
| FITC anti-human TNF-α, Clone Mab11 | Biolegend | Cat. #502906, 1:50 |
| PE/Dazzle™ 594 anti-human IFN-γ Antibody, Clone 4S.B3 | Biolegend | Cat. #502546, 1:50 |
| BiP (C50B12) Rabbit mAb, | Cell Signalling | Cat. #3177, 1:100 (FC), 1:1000 (WB) |
| Alexa Fluor® 647 anti-human FOXP3 Antibody | Biolegend | Cat.#320114, 1:100 |
| FITC anti-human CD45RA Antibody | Biolegend | Cat. #304106, 1:100 |
| PE/Dazzle™ 594 anti-human CD57 Antibody | Biolegend | Cat. #359620, 1:100 |
| Alexa Fluor® 647 anti-Puromycin Antibody | Biolegend | Cat. #381507, 1:400 |
| Goat anti-Rabbit IgG (H + L) Highly Cross-Adsorbed Secondary Antibody, Alexa Fluor™ 594 | ThermoFisher | Cat. #A-11037, 1:200 |
| Goat anti-Mouse IgG (H + L) Cross-Adsorbed Secondary Antibody, Alexa Fluor™ 488 | ThermoFisher | Cat. #A-11001, 1:200 |
| IRE1α (14C10) Rabbit mAb | Cell Signaling | Cat. #3294, 1:100 |
| PERK (D11A8) Rabbit mAb | Cell Signaling | Cat. #5683, 1:100 |
| ATF-4 (D4B8) Rabbit mAb | Cell Signaling | Cat. #11815, 1:100 (FC) 1:1000 (WB) |
| ATF5 Monoclonal antibody | Proteintech | Cat. #67066-1-1 g, 1:100 |
| SQSTM/p62 antibody (D-3) | Santa Cruz Biotechnology | Cat. # sc-28359, 1:100 |

| Reagent/resource | Reference or source | Identifier or catalog number |
|---|---|---|
| XBP-1s (D2C1F) Rabbit mAb | Cell Signaling | Cat. #12782, 1:1000 |
| PINK1 (D8G3) Rabbit mAb | Cell Signaling | Cat. #6946S, 1:2000 |
| Alexa Fluor® 647 anti-human CD25 Antibody | Biolegend | Cat. #356128, 1:75 (imaging) |
| Alexa Fluor® 647 anti-human CD56 Antibody | Biolegend | Cat. #362513, 1:50 (imaging) |
| Pacific Blue™ anti-human CD16 Antibody | Biolegend | Cat. #302024, 1:50 (imaging) |
| Alexa Fluor® 647 anti-human CD16 Antibody | Biolegend | Cat. #302020, 1:50 (imaging) |
| Pacific Blue™ anti-human CD56 (NCAM) Antibody | Biolegend | Cat. #362520, 1:50 (imaging) |
| Pacific Blue™ anti-human CD127 (IL-7Rα) Antibody | Biolegend | Cat. #351306, 1:50 (imaging) |
| Goat anti-Rabbit IgG (H + L) Cross-Adsorbed Secondary Antibody, Alexa Fluor™ 488 | ThermoFisher | Cat. #A-11008, 1:200 |
| Goat anti-Rabbit IgG (H + L) Highly Cross-Adsorbed Secondary Antibody, Alexa Fluor™ 594 | ThermoFisher | Cat. #A-11037, 1:200 |
| Anti-mouse IgG (H + L), HRP-conjugate | Promega | Cat. #W4021, 1:10,000 |
| Anti-rabbit IgG, HRP-linked Antibody | Cell Signaling | Cat. #7074, 1:10,000 |
| **Oligonucleotides and other sequence-based reagents** | | |
| Not applicable | | |
| **Chemicals, enzymes, and other reagents** | | |
| ER tracker red | ThermoFisher | Cat. #E34250 |
| Mitotracker green | ThermoFisher | Cat. #M46750 |
| Image-iT™ TMRM Reagent | ThermoFisher | Cat. #I34361 |
| MitoSOX™ Mitochondrial Superoxide Indicators | ThermoFisher | Cat. #M36007 |
| APO-BrdU™ TUNEL Assay Kit, with Alexa Fluor™ 488 Anti-BrdU | ThermoFisher | Cat. #A23210 |
| PROTEOSTAT® Aggresome detection kit | Enzo Lifesciences | Cat. #ENZ-51035-0025 |
| Tunicamycin | Sigma | Cat. #T7765 |
| TBHP | ThermoFisher | Cat. #C10492 |
| Oxidized Low-Density Lipoprotein (OxLDL) | ThermoFisher | Cat. #L34357 |
| Human IFN-gamma Recombinant Protein, PeproTech® | ThermoFisher | Cat.#300-02-20UG |
| Human IL-2 Recombinant Protein, PeproTech® | ThermoFisher | Cat.#200-02-50UG |
| Amersham™ Protran® Western blotting membranes, nitrocellulose | Amersham | Cat.#GE10600001 |

| Reagent/resource | Reference or source | Identifier or catalog number |
|---|---|---|
| Bovine serum albumin | Cell Signaling | Cat. #9998S |
| Blotting Grade Blocker Non-Fat Dry Milk | Biorad | Cat. #1706404XTU |
| TGX™ FastCast™ Acrylamide Kit, 12% | Biorad | Cat. #1610175 |
| Ammonium persulfate | Sigma | Cat. #A3678 |
| TEMED | Biorad | Cat. #1610800 |
| RIPA Lysis buffer | Cell Signaling | Cat. #9806 |
| Protease Inhibitor | Roche | Cat.#11697498001 |
| 4X Laemmli sample buffer | Biorad | Cat.#1610747 |
| 2-Mercaptoethanol | ThermoFisher | Cat.#21985023 |
| SuperSignal™ West Femto Maximum Sensitivity Substrate 100 ml | ThermoFisher | Cat.#34095 |
| Dynabeads™ Human T-Activator CD3/CD28 for T Cell Expansion and Activation | ThermoFisher | Cat.#11161D |
| TexMACS | Miltenyi Biotec | Cat.#130-097-196 |
| RPMI 1640 Medium, no phenol red | ThermoFisher | Cat.#11835030 |
| Puromycin | Sigma | Cat.#P7255 |
| 2-Deoxy-D-Glucose | Sigma | Cat.#D6134 |
| Oligomycin A | Sigma | Cat.#75351 |
| eBioscience™ Fixation/ Permeabilization Concentrate | ThermoFisher | Cat.#00-5123-43 |
| eBioscience™ Fixation/ Permeabilization Diluent | ThermoFisher | Cat.#00-5223-56 |
| eBioscience™ Permeabilization Buffer (10X) | ThermoFisher | Cat.#00-8333-56 |
| Glass-bottom dishes | Cellvis | Cat.#D35-20-1.5-N |
| Poly D lysine | ThermoFisher | Cat.#A3890401 |
| EasySep™ Human CD4 + T Cell Isolation Kit | STEMCELL Technologies | Cat.#17952 |
| EasySep™ Human Naïve CD4 + T Cell Isolation Kit | STEMCELL Technologies | Cat.#19555 |
| EasySep™ Human CD4+CD127lowCD25+ Regulatory T Cell Isolation Kit | STEMCELL Technologies | Cat.#18063 |
| ImmunoCult™-XF T Cell Expansion Medium | STEMCELL Technologies | Cat.#10981 |
| ImmunoCult™ Human Treg Differentiation Supplement | STEMCELL Technologies | Cat.#10977 |
| 1×Cell Stimulation Cocktail, eBioscience | ThermoFisher | Cat.# 00-4970-93 |
| 1× Protein Transport Inhibitor Cocktail, eBioscience | ThermoFisher | Cat.# 00-4980-93 |
| **Software** | | |
| FlowJo version 10.10.0. | Becton Dickinson (BD) | |

| Reagent/resource | Reference or source | Identifier or catalog number |
|---|---|---|
| FIJI is just ImageJ | ImageJ, NIH | |
| GraphPad Prism version 10 | GraphPad | |
| Gene set enrichment analysis | UC San Diego/Broad Institute | https://www.gsea-msigdb.org/gsea/index.jsp |
| **Other** | | |
| Cytek Aurora 5-Laser Spectral Cytometer | Cytek Biosciences | |
| Nikon AX R with NSPARC confocal system | Nikon | |

## Human samples

Healthy human donors were recruited via the clinical core at the La Jolla Institute for Immunology (LJI), La Jolla, California, USA. Written informed consents were obtained from all donors, and they were financially compensated according to the guidelines approved by LJI's institutional review board (IRB). Donors tested negative for any significant systemic disease or infections, including hepatitis B or C and HIV. Ethical approval for the study was provided by LJI's IRB (protocol no. VD-057). Demographic details are listed in Tables 1 and 2.

De-identified cryopreserved PBMCs were obtained from CAD subjects undergoing standard cardiac catheterization at the University of Virginia Health System, Charlottesville, Virginia, USA [Coronary Assessment in Virginia (CAVA) cohort]. All participants provided written informed consents before enrollment. Approval of the study was obtained from the Human Institutional Review Board (IRB No. 15328) at the University of Virginia. Demographic details are listed in Table 2.

The experiments conformed to the principles set out in the WMA declaration of Helsinki and the Department of Health and Human Services' Belmont Report. Sample sizes were based on previous experience, providing enough statistical robustness and reproducibility, and were based on the availability of resources. No data were excluded from the analysis.

## Isolation of human PBMCs

Human blood samples collected in EDTA-coated tubes were centrifuged at 800 ×g for 15 min with no brakes at 24 °C. After removing the plasma layer on top, an equal amount of serum-free cell culture medium (TexMACS, Miltenyi Biotec, 130-097-196) was added and thoroughly mixed. The diluted sample was carefully layered on top of Ficoll-Paque Plus (Millipore Sigma, 17-1440-02) in the ratio of 7:3. Samples were centrifuged at 800 ×g for 30 min with brakes off at 24 °C, and the layer of PBMCs at the interface was transferred into a fresh tube. Cells were washed twice with 1× phosphate-buffered saline (PBS; w/o Ca/Mg, Gibco, 10010023) at 800 ×g, 10 min. The red blood cells and platelets were removed using RBC lysis buffer (Thermo Fisher Scientific, 00-4333-57) for 5 min followed by centrifugation at 250 ×g for 10 min. Cells were counted with a hemocytometer, and Trypan Blue was used to

**Table 1. Summary of demographic characteristics of healthy subjects.**

| Age (years) | 25–55, (median = 35) |
|---|---|
| Male | 8 (61.5%) |
| Female | 5 (38.5%) |

**Table 2. Summary of demographic and clinical characteristics of subjects used in Fig. 4C–F.**

| | Healthy (n = 7) | CAD (n = 9) |
|---|---|---|
| Age (years) | 52–69, (median = 60) | 51–77, (median = 74) |
| Male | 5 (71.4%) | 5 (55.5%) |
| Female | 2 (28.5%) | 4 (44.4%) |
| Diabetes (yes) | 0 | 4 (44.4%) |
| Gensini score | Not applicable | 3.5–45 (median = 30.5) |

determine viability. PBMCs were resuspended in CryoStor® CS10 (STEMCELL, 07930) and cryopreserved in liquid nitrogen.

## In vitro culture of human PBMC's

PBMCs were thawed as described in the previous section and then resuspended in TexMACS medium supplemented with 1% penicillin/streptomycin (Thermo Fisher Scientific, 15140122). Cells were plated at a density of $1.5 \times 10^6$ cells/ml in 48-well plates and cultured at 37 °C/5% $CO_2$ in a humidified incubator.

For inducing ER stress, cells were treated with 25 µM tunicamycin (Sigma, T7765) for 72 h. For inducing mitochondrial depolarization, cells were treated with 400 µM TBHP (Thermo Fisher Scientific, C10492) for 24 h. In tunicamycin, 5–10% of CD3 + T cells died after treatment. For TBHP, 20% of CD3 + T cells died post-treatment.

For oxLDL and IFNγ experiments, cells were treated with 20 µg/ml oxLDL (Thermo Fisher Scientific, L34357) or 10-20 ng/ml IFNγ (Peprotech, Thermo Fisher Scientific, 300-02-20UG) for 72 h. Untreated and treated cells were subsequently analyzed for Treg and exTreg frequency or uptake of various dyes by flow cytometry as described in the previous section.

## In vitro generation of iTregs

CD3 + CD4 + CD8− CD45RA + CD45RO− naive T cells and CD3 + CD4 + CD8- CD45RA-CD45RO+ memory T cells were sorted on ThermoFisher-Big foot cell sorter. For some experiments, naïve T cells were enriched using EasySep™ Human Naive CD4 + T Cell Isolation Kit (Stem Cell Technologies, 19555). Similar downstream results were obtained with both ways of isolation. Sorted or enriched Naive T cells ($1 \times 10^6$ cells/mL) were rested overnight in Immunocult T Cell Expansion Medium (Stem Cell Technologies, 10981) supplemented with Immuno-cult Treg differentiation supplement (Stem Cell Technologies, 10977) as per the manufacturer's instructions and incubated at 37 °C and 5% $CO_2$. The next day, the cells were activated using Dynabeads™ Human T-Activator CD3/CD28 (ThermoFisher, 11161D). Cell density was adjusted to $1 \times 10^6$ cells/mL every few days as required by adding fresh complete Treg Differentiation Medium. On Day 7, differentiated iTregs

were checked by flow cytometry for expression of CD25 and FOXP3 and used for downstream experiments as indicated in the figure legends.

## Flow cytometry

Cryopreserved PBMCs were thawed at 37 °C in a water bath and subsequently washed with PBS (w/o Ca/Mg) by centrifugation at $400 \times g$, 10 min. Cells were counted using a hemocytometer, and viability was determined using the Trypan Blue dye exclusion method. Cells were then washed with cold FACS buffer [PBS w/o Ca/Mg, 2% fetal bovine serum (FBS)] at $400 \times g$, 5 min, 4 °C. For surface markers, cells were stained for 40 min at 4 °C with antibodies listed in the Reagents and tools table. Cells were washed with cold FACS buffer and analyzed by flow cytometry.

For staining with various organelle/functional dyes following surface marker staining, cells were incubated with 100 nM ER tracker red (Thermo Fisher Scientific, E34250) to label the ER. For mitochondrial labeling, cells were incubated with 100 nM TMRM (Thermo Fisher Scientific, I34361) or 100 nM Mitotracker green (Thermo Fisher Scientific, M46750). Mito-chondrial ROS were determined by staining the cells with 500 nM MitoSox Red (Thermo Fisher Scientific, M36007). For apoptosis detection, cells were incubated with Annexin V (Biolegend, 640912) at 1:100 dilution. All these incubations were done in Hank's Balanced Salt Solution with calcium and magnesium (HBSS/Ca/Mg, Gibco, 14025-092) for 30 min at 37 °C/5% $CO_2$. For the TUNEL assay, cells were stained using the manufacturer's instructions (Thermo Fisher Scientific, A23210). For studying the accumulation of misfolded proteins, cells were stained with Proteostat as per the manufacturer's instructions (Enzo Lifesciences, ENZ-51023). Cells were finally washed, resuspended in FACS buffer, and data were acquired on Cytek Aurora 5-Laser Spectral Cytometer. Voltages were set up using single color-stained cells and compensation beads (Invitrogen, 01-2222-42). Data were analyzed using FlowJo version 10.10.0.

## Intracellular antibody staining

For staining of intracellular proteins FOXP3, BiP, CHOP, IRE1, PERK, P62, ATF4, and ATF5, following surface staining, cells were fixed in eBioscience Foxp3/Transcription factor staining buffer (ThermoFisher, 00-5523-00) and incubated with the indicated antibodies (details in reagents and tools table) in 1X permeabilization buffer for 40 min at RT. Apart from FOXP3, for all other primary antibodies, cells were subsequently incubated with relevant Alexa 488/Alexa 594 secondary antibodies (details in reagents and tools table) for 30 min at 4 °C. Finally, cells were washed and resuspended in FACS buffer for analysis by flow cytometry using Cytek Aurora 5-Laser Spectral Cytometer. The data for Tregs and exTregs were plotted after normalizing to isotype control staining in respective cells.

## Intracellular cytokine staining

For the intracellular cytokine staining assay, human PBMC's were stimulated for 6 h with PMA and ionomycin (1× Cell Stimulation Cocktail, eBioscience, 00-4970-93). During the last 4 h, protein transport inhibitors brefeldin and monensin (1× Protein Transport Inhibitor Cocktail, eBioscience, 00-4980-93) were added. Following staining with viability and surface markers as described in the previous section, cells were fixed and permeabilized in fixation buffer (eBioscience IC Fixation buffer, 00-8222-49) followed by staining with antibodies against IFNg or TNF for 40 min at 24 °C and analyzed on Cytek Aurora 5-Laser Spectral Cytometer.

## Metabolic profiling by SCENITH

Metablic profiling of Tregs and exTregs was done by SCENITH (Arguello et al, 2020). PBMCs were thawed as described earlier and $1 \times 10^6$ cells were seeded in 0.5 ml media in a 48-well plate in RPMI + 10% FBS (without antibiotics). After resting these cells for 1 h, wells were treated with DMSO control, 2-deoxy-D-Glucose (DG, final concentration of 100 mM), Oligomycin (Oligo, final concentration of 1 μM), or a sequential combination of these drugs (DGO) at the final concentrations mentioned before for 30–45 min. Puromycin (final concentration of 10 μg/ml) was added during last 30 min of the treatment with metabolic inhibitors. After that, the, the samples were washed with cold FACS buffer and stained as described previously. Intracellular staining for Puromycin was done as described in the intracellular antibody section. Staining was done for 50 min on ice at the antibody dilution of 1:400. Eventually, cells were washed and analyzed on Cytek Aurora 5-Laser Spectral Cytometer. Mitochondrial dependence was calculated using the following formula: $100 \times$ (MFI of puromycin in control-MFI of puromycin in Oligo)/(MFI of puromycin in control-MFI of puromycin in DGO).

## Western blotting

After treatment of iTregs with various stressors as described in the figure legends, $1 \times 10^6$ iTregs were lysed on ice (with intermittent vortexing) for 30 min in RIPA buffer (Cell Signaling Technology; 9806) containing 2× protease inhibitor (Roche; 11697498001; Basel, Switzerland). The lysate was heated at 65 °C for 10 min in 1× Laemmli buffer on a thermal mixer-shaker and fractionated on an 12% SDS polyacrylamide gel for immunoblotting. Following protein transfer on nitrocellulose membrane (Amersham, #GE10600001), membranes were blocked in 5% BSA or 5% skim milk for 1 h at RT. The following primary antibodies were used; anti-BiP (Cell Signaling, #3177, 1:1000), anti-CHOP (Cell Signaling, #2895, 1:1000), anti-β-actin (Cell Signaling, #4967, 1:5000), anti-PINK1 (Cell signaling, # 6946S, 1:2000), anti-sXBP1 (Cell Signaling,# 12782S, 1:1000) and anti-ATF4 (Cell Signaling, #11815S, 1:1000).

All primary incubations were done overnight at 4 °C. The following secondary antibodies were used: anti-mouse IgG (H + L), HRP-conjugate (Promega, #W4021; 1:10,000), and anti-rabbit IgG, HRP-linked antibody (Cell Signaling, #7074, 1:10,000). Secondary incubations were done at RT for 1 h.

## Fluorescence microscopy

For the live-cell imaging experiments, Tregs were cultured on glass-bottom dishes (Cellvis; D35-20-1.5-N) after coating them with Poly-D-Lysine (ThermoFisher, A3890401). For mitochondrial labeling, untreated and tunicamycin-treated cells were incubated with Mitotracker green (100 nM) in TexMACS medium for 30 min at 37 °C/5% $CO_2$. Subsequently, cells were washed twice with PBS, pH 7.4, and imaged in the imaging medium; RPMI without phenol red (ThermoFisher, #11835030) + 10% fetal bovine serum. Images were captured on Nikon AX R with NSPARC confocal system with 60× Plan-Apochromat 1.4(oil) objective NA 1.42 (MRD71670), equipped with 4-line (405 nm, 488 nm, 561 nm, and 647 nm) laser and Prime 95B back-thinned sCMOS camera (Teledyne Photometrics). Images were captured using AX 25 mm FOV galvano scanner and a scan area of 256 × 256 pixels at a Nyquist resolution of 0.146 μm. Cells were imaged while in a stage-top environmental

chamber (Tokai Hit) and running NIS Elements software. Images were deconvolved using the automatic 2D deconvolution feature, point scan confocal modality with preprocessing setting of "do not subtract" and automatic computation of PSF.

For analyzing mitochondrial morphology, single-channel images were subjected to unbiased auto-thresholding using the Yen algorithm on ImageJ to generate binary black-and-white images with black objects showing the labeling pattern for mitotracker. Analyze Particles command was then used to generate a mask of the images (size criteria >0.2 μm² and circularity criteria of 0–0.3). These criteria clearly identified cells having elongated mitochondrial tubules. To calculate % of cells showing elongated mitochondrial morphology, the number of cells that passed the above criteria were divided by the total number of cells that were analyzed, and the resulting ratio was multiplied by 100. The results were also confirmed by manual scoring of blinded images by two independent researchers.

## Differential expression of genes in Tregs vs exTregs

Volcano plots to visualize DE genes and scaled heatmaps of normalized gene expression values (transcripts per million) were made using tools available at https://www.bioinformatics.com.cn/en, a free online platform for data analysis and visualization. Gene set enrichment analysis was done with GSEA (v4.3.2) using gene sets in Human mSigDB. Bar graphs for comparing normalized gene expression values (transcripts per million) in Tregs vs exTregs were generated using GraphPad Prism (v10.0.0).

## Analysis of human scRNA-seq data from blood and plaque

Processing of publicly available dataset (GSE196943) was performed using Seurat (v5) in R. The following cells were normalized to low-quality and excluded; cells with <300 detected genes, RNA counts >20,000, or mitochondrial transcript percentages >10%. Doublet detection was done using scDblFinder. Each Seurat object was transformed into a Single-Cell Experiment (SCE) object, and doublet identification and removal were done based on classification scores. Filtered data were normalized using SCTransform (SCT) to regress out mitochondrial effects. PCA and ElbowPlot were used to determine the optimum number of dimensions for downstream clustering. A shared nearest-neighbor (SNN) graph was made using the FindNeighbors function, and subsequent clustering was done with FindClusters at a resolution of 0.5. To correct batch effects across various samples, an integration workflow was applied. SelectIntegrationFeatures was used to identify substantially variable genes across samples, and FindIntegrationAnchors was used to compute batch correction anchors. Samples were finally integrated using IntegrateData. Cell types were determined with SingleR using Human Primary Cell Atlas reference. CD4 + T cells were identified based on expression of the following markers; CD3E + , CD4 + , CD8A−, CD19−, CD14−, and embedded on a UMAP. Sample-based expression of various genes was analyzed using AverageExpression.

## Statistical analysis

Data analysis and statistical comparisons were done using GraphPad Prism version 10. Comparison of expression of different genes in Tregs vs

## The paper explained

### Problem

Regulatory T cells (Tregs) keep autoimmune responses in check by suppressing the function of effector (killer) T cells. Treg dysfunction and/or deficiency can cause autoimmune and inflammatory diseases like coronary artery disease (CAD). In such diseases, the inflammatory environment makes Tregs unstable, resulting in loss of their suppressive function. The cellular mechanisms driving this instability are not well understood.

### Results

In our study, we found that endoplasmic reticulum (ER) stress triggers Treg instability by causing mitochondrial dysfunction. Human exTregs, a recently identified subset of Treg-derived cells, exhibit ER stress and disturbed mitochondrial homeostasis. Activation of the integrated stress response (ISR) in exTregs induces an inflammatory phenotype. Physiological stressors implicated in CAD can induce ER stress and mitochondrial dysfunction in Tregs in vitro. Consistent with this, Tregs from CAD patients show high ER stress and low mitochondrial membrane potential, traits that we show are typical of dysfunctional exTregs.

### Impact

Therapeutic interventions promoting resolution of ER stress or ameliorating mitochondrial function may promote Treg function in CAD and other inflammatory diseases.

---

exTregs in human bulk RNA sequencing data was done using a two-tailed Mann–Whitney test. Two-sample comparisons were done with a two-tailed paired Student's *t* test. Multiple sample comparisons were done with one-way or two-way ANOVA. Details of the statistical tests used for each experiment are listed in the figure legends, and numerical *P* values are listed at the top of each bar graph.

## Data availability

This study includes no data deposited in external repositories.

The source data of this paper are collected in the following database record: biostudies:S-SCDT-10_1038-S44321-025-00322-3.

## Peer review information

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

## Acknowledgements

We acknowledge the support from the National Heart, Lung and Blood Institute (awards P01 HL136275 and R35 HL145241) to KL. We thank the Augusta University (AU) Georgia Cancer Center Flow and Mass Cytometry Core Facility (RRID: SCR_025747) for the acquisition of flow cytometry data. We also thank Augusta University Medical College of Georgia Cell Imaging Core Facility (RRID:SCR_026799) for the acquisition of confocal imaging data. We thank members at the cardiac catheterization laboratories at the University of Virginia (UVA). We thank Melissa Marshal from UVA and Michael Nakama from the Immunology Center of Georgia for coordinating the maintenance and transport of these samples. We also thank the Clinical Core at La Jolla Institute for Immunology for the collection of human blood samples. The graphical abstract was generated using graphics from Servier Medical Art.

## Author contributions

**Smriti Parashar**: Conceptualization; Data curation; Formal analysis; Supervision; Validation; Investigation; Visualization; Methodology; Writing—original draft; Project administration; Writing—review and editing. **Mohammad Oliaeimotlagh**: Resources; Formal analysis; Investigation; Methodology. **Payel Roy**: Resources; Formal analysis; Investigation; Methodology. **Qingkang Lyu**: Investigation; Methodology. **Anusha Bellapu**: Investigation. **Mikhail Fomin**: Investigation. **Sunil Kumar**: Investigation. **Yan Wang**: Investigation. **Chantel C McSkimming**: Resources. **Coleen A McNamara**: Resources; Project administration; CAM provided clinical expertise, patient samples and medical data. **Klaus Ley**: Conceptualization; Supervision; Funding acquisition; Writing—original draft; Project administration; Writing—review and editing.

Source data underlying figure panels in this paper may have individual authorship assigned. Where available, figure panel/source data authorship is listed in the following database record: biostudies:S-SCDT-10_1038-S44321-025-00322-3.

## Disclosure and competing interests statement

The authors declare no competing interests.

# Expanded View Figures

**Figure EV1.  ER stress in human Tregs and exTregs.**

(**A**) Normalized expression levels (transcripts per million) of ER-stress genes *XBP1, ATF6B* and *HSPA5* in human bulk transcriptomes from sorted human Tregs and exTregs. Horizontal bars represent the median. $n = 7$. (**B**) Gating strategy for identifying Tregs (CD3 + CD4 + CD8-CD25+CD127lo) and exTregs (CD3 + CD4 + CD8-CD25-CD56 + CD16 + ) in human PBMC's. (**C–E**) Representative histograms for intracellular staining of IRE1 (**C**), PERK (**D**) and CHOP (**E**) by flow cytometry. Tregs (blue), exTregs (red) and isotype control (gray) are shown. The *y* axis was normalized to mode. (**F**) Representative histograms for intracellular staining of Proteostat by flow cytometry. Tregs (blue), Tregs + MG132 (purple) and unstained cells (gray) are shown. The *y* axis was normalized to mode. (**G**) Flow cytometry plots showing generation of iTregs by inducing CD3 + CD4 + CD45RA + CD45RA- Naïve T cells with ImmunoCult™ Human Treg Differentiation Supplement (containing TGFβ1 and all-trans retinoic acid) and Dynabeads human CD3/CD28 activator for 7 days (Left). Memory T cells (CD3 + CD4 + CD45RA-CD45RA + ) were used as control (Right). (**H**) Representative flow plots showing frequency of CD25 + FOXP3+ Tregs in untreated vs Tunicamycin (72 h) treated PBMC's. (**I**) Bar graph showing frequency of CD25+CD127lo Tregs in PBMC's treated with Tunicamycin for 72 h. Untreated cells were used as control. Each dot represents a biological replicate from an independent human donor. $n = 6$. Representative flow plots are shown on the right. (**J**) natural Tregs (nTregs) or induced Tregs (iTregs) from same donors were treated with Tunicamycin for 24 h and relative MFI of FOXP3 was plotted against untreated controls from the respective donor. MFI of FOXP3 in untreated control from each donor was normalized to 100. $n = 3$. Each dot represents a biological replicate from an independent human donor. Statistical comparisons were done using two-tailed Mann–Whitney *U* test in (**A**) and using paired two-tailed *T* test in (**I, J**). Results are represented as mean ± SEM. Numerical *P* values are listed at the top of each bar graph. Source data are available online for this figure.

▶

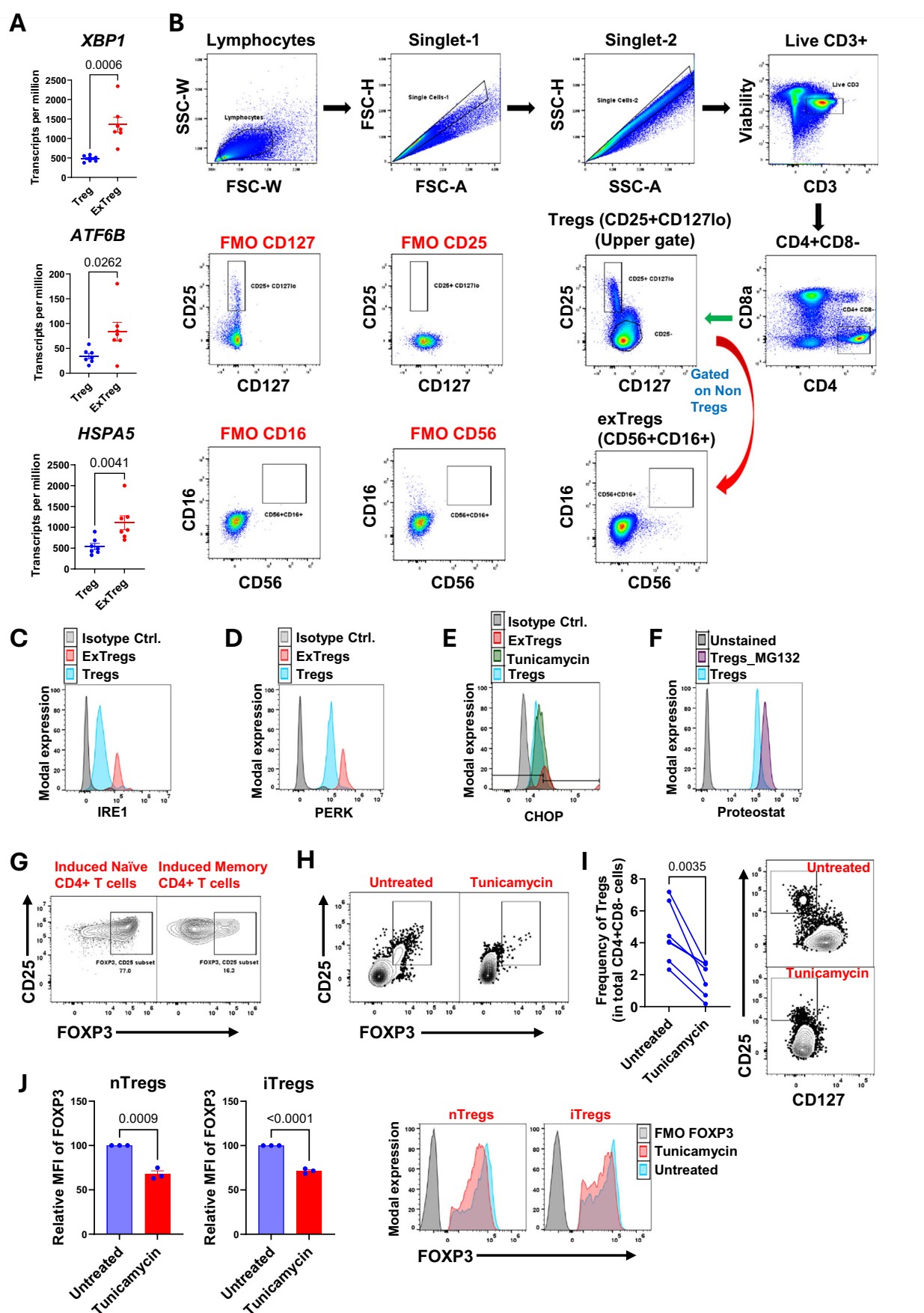

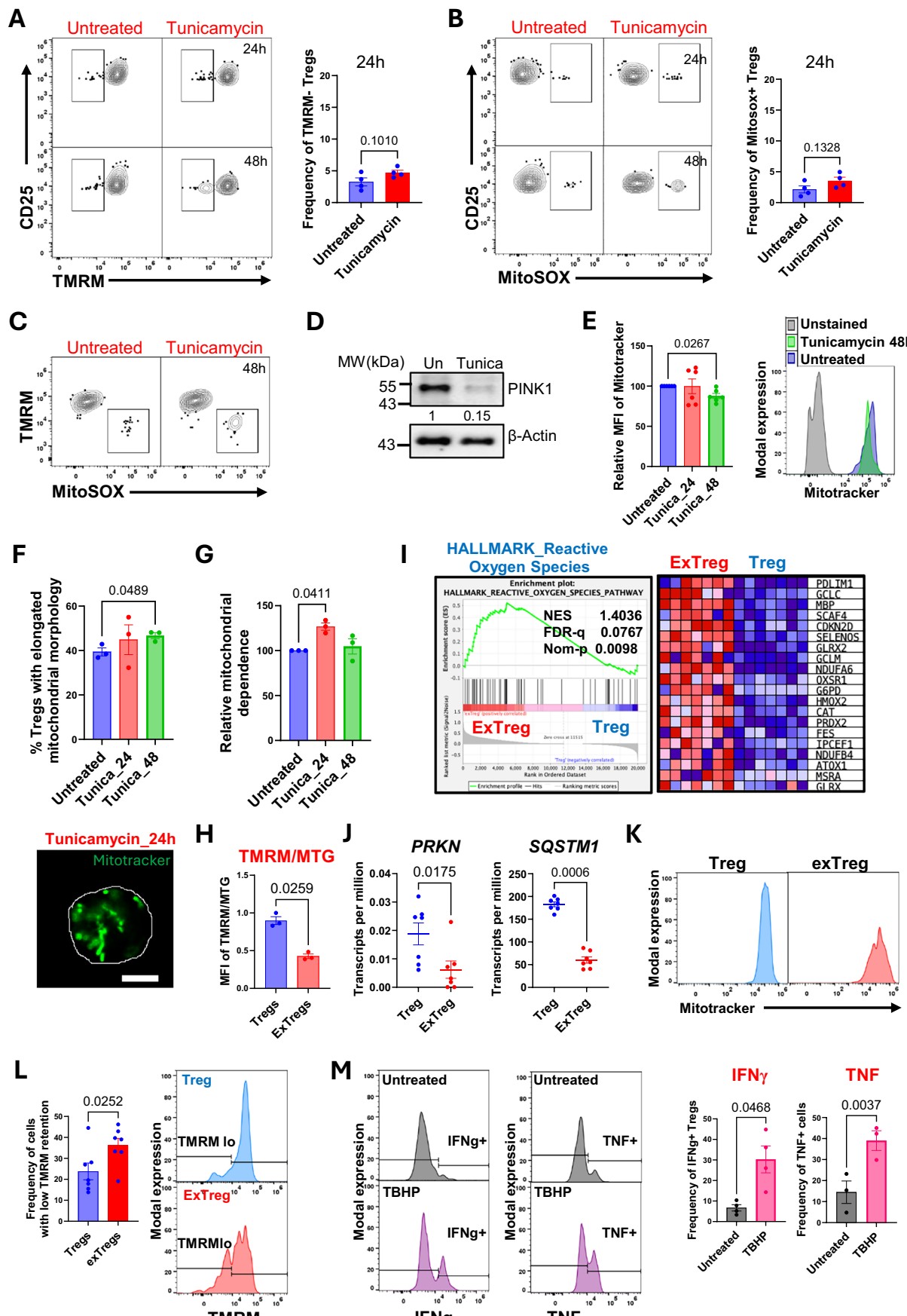

◄ **Figure EV2. Tunicamycin and oxidative stress.**

(A, B) Representative flow cytometry plots showing the frequency of TMRM− (A) or Mitosox+ (B) Tregs in PBMCs treated with tunicamycin for 24 h or 48 h. Untreated cells were used as control. Bar graph on the right of each plot shows the frequency of TMRM− (A) or Mitosox+ (B) Tregs after 24 h of tunicamycin treatment. Each dot represents a biological replicate from an independent human donor. $n = 4$. (C) Representative flow cytometry plots showing the frequency of Tregs that are TMRM- and Mitosox+ in untreated and tunicamycin treated samples. (D) In vitro generated iTregs were treated with tunicamycin for 24 h and the protein expression of PINK1 was analyzed by western blotting. Molecular weights (kDa) are indicated on the left of the blots. β Actin was used as loading control. Normalized levels of PINK1 to β Actin are indicated at the bottom of the blot. (E) Bar graph comparing the MFI of mitotracker in Tregs from PBMC's treated with tunicamycin for 24 h and 48 h. Untreated PBMC'S were used as control. Right, Representative histograms showing the fluorescence intensity of mitotracker in Tregs from untreated (blue) and tunicamycin treated (green) PBMC's. Unstained cells are shown in gray. The y axis was normalized to mode. (F) Tregs were cultured in vitro and treated with tunicamycin for 24 h and 48 h. Mitochondria were labeled with mitotracker green and visualized by live-cell confocal imaging. Untreated Tregs were used as control. Bar graph shows % of Tregs that showed tubular/elongated morphology. Each dot represents a biological replicate from an independent human donor. $n = 3$. Representative image for 24 h-tunicamycin treated samples are shown at the bottom. Scale bar, 5 μm. 15–30 cells were analyzed from each donor for each condition. (G) Human PBMC's were treated with tunicamycin for 24 h and 48 h and mitochondrial dependence of Tregs was determined using SCENITH. Bar graph shows relative mitochondrial dependence in tunicamycin treated cells compared to untreated control. Mitochondrial dependence in untreated control from each donor was set to 100. Each dot represents a biological replicate from an independent human donor. $n = 3$. (H) Human PBMCs were stained with TMRM and Mitotracker green and analyzed by flow cytometry. MFI of TMRM/MTG in Tregs from each donor was calculated and MFI of TMRM/MTG for exTregs from respective donors was plotted against it. Each dot represents a biological replicate from an independent human donor. $n = 3$. (I) GSEA plots showing enrichment of Hallmark gene signature for Reactive Oxygen Species (M5938 in mSigDB) within exTreg and Treg ($n = 7$) transcriptomes. Normalized enrichment score (NES), FDR q and nominal p values are indicated. Heatmap for top 10 enriched genes in exTregs vs Tregs is shown at bottom. Color scale in the heatmap is based on GSEA row minimum (blue) to row maximum (red). (J) Normalized expression levels (transcripts per million) of mitophagy genes *PRKN* and *SQSTM1* in human bulk transcriptomes from sorted human Tregs and exTregs. Horizontal bars represent the median. $n = 7$. (K) Representative flow cytometry plots showing mitotracker staining in Tregs (blue) and exTregs (red). y axis was normalized to mode. (L) Human PBMC's were stained with TMRM and analyzed by flow cytometry. Bar graph on left depicts the frequency of Tregs and exTregs that show low TMRM retention. Representative flow cytometry plots for Tregs (blue) and exTregs (red) are shown on the right. y axis was normalized to mode. (M) Human PBMC's were treated with TBHP and frequency of cells that express IFN γ or TNF was analyzed by flow cytometry. Untreated cells were used as control. Representative flow cytometry plots for each cytokine (untreated in gray and TBHP treated in purple) are shown on the left. Bar graphs comparing the frequency are shown on the right. $n = 4$ for IFN γ and $n = 3$ for TNF. Each dot represents a biological replicate from an independent human donor. Statistical comparisons were done using paired two-tailed T test in (A, B, H, L, M), one-way ANOVA with Tukey's multiple comparisons in (E–G) and two-tailed Mann–Whitney U test in (J). Results are represented as mean ± SEM. Numerical P values are listed at the top of each bar graph. Source data are available online for this figure.

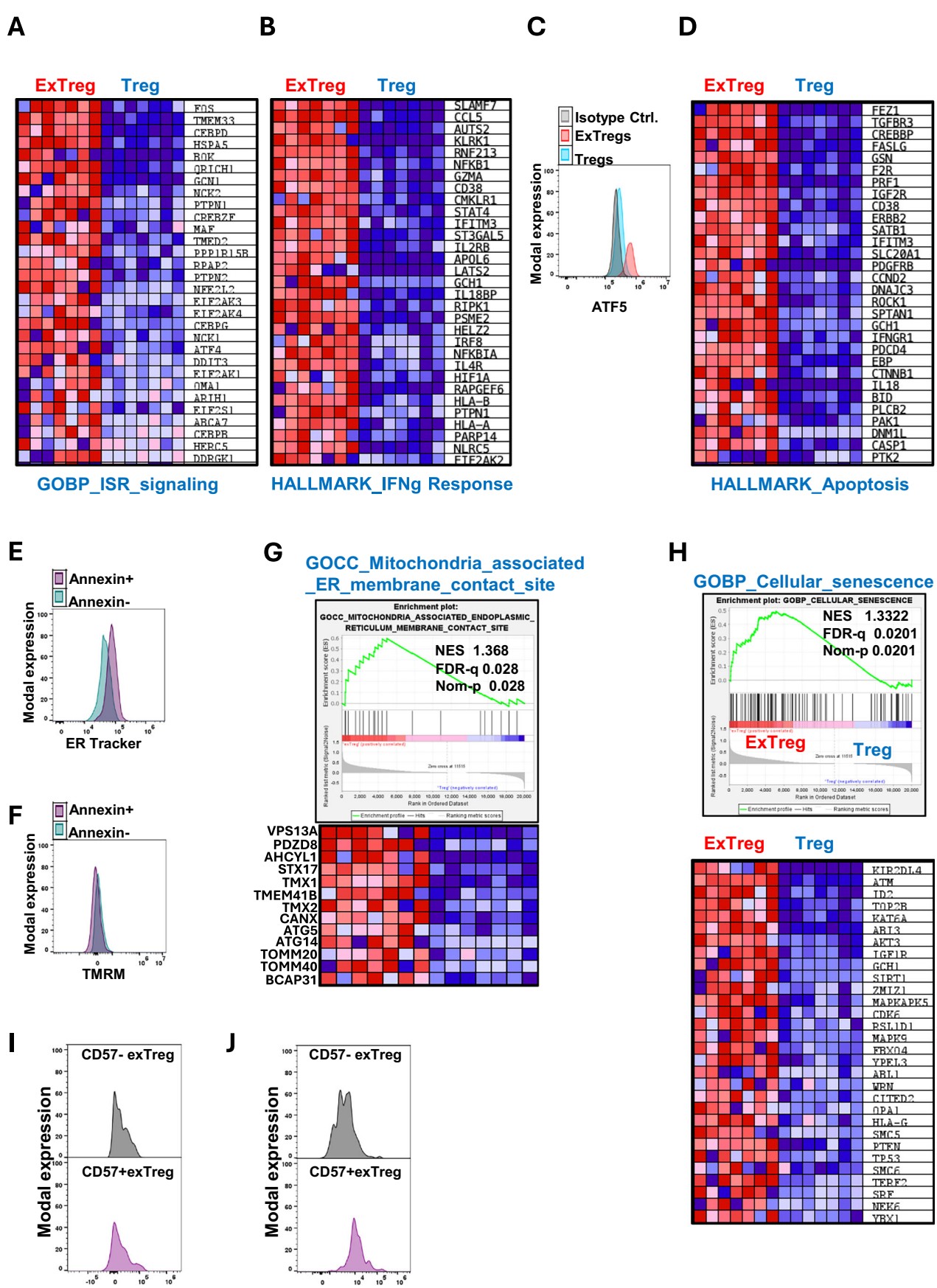

**Figure EV3. Gene set enrichment for integrated stress response and interferon gamma response.**

(A, B) Heatmaps showing top 30 enriched genes in exTreg vs Treg transcriptomes as revealed by GSEA for I\integrated stress response signaling (A) and interferon gamma response (B). Color scale in the heat map is based on GSEA row minimum (blue) to row maximum (red). (C) Representative flow cytometry plot for ATF5 staining in Tregs (blue), exTregs (red) and isotype control (gray). Y axis was normalized to mode. (D) Heatmap showing top 30 enriched genes in exTreg vs Treg transcriptomes as revealed by GSEA for Apoptosis. Color scale in the heat map is based on GSEA row minimum (blue) to row maximum (red). (E, F) Representative flow cytometry plots comparing ER tracker (E) and TMRM (F) staining in Annexin+ (purple) and Annexin- (turquoise) exTregs. Y axis was normalized to mode. (G, H) GSEA plots showing enrichment of gene signature for mitochondria-associated ER membrane contact sites (GO:0044233 in mSigDB) in (G) and cellular senescence (GO:0090398) in (H) within paired exTreg and Treg ($n = 7$) transcriptomes. Normalized enrichment score (NES), FDR q and Nominal $P$ values are indicated. Top enriched genes in exTregs are shown at the bottom. Color scale in the heat map is based on GSEA row minimum (blue) to row maximum (red). (I, J) Representative flow cytometry plots comparing TMRM (I) and Annexin V (J) staining in CD57− (gray) and CD57+ (purple) exTregs. Y axis was normalized to mode.

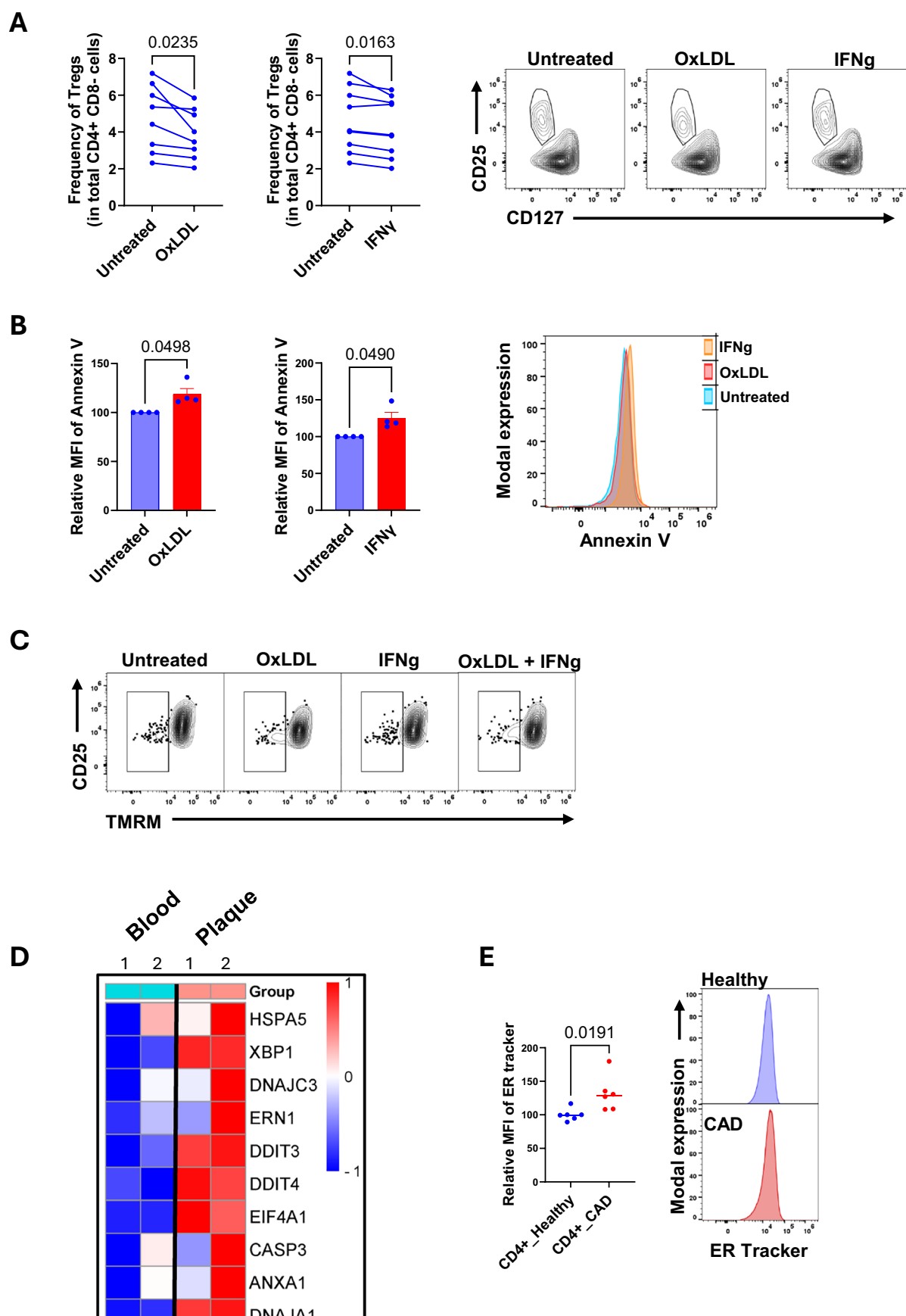

◀ **Figure EV4. oxLDL and IFNγ treatment.**

(A) Human PBMCs were treated with oxLDL or IFN γ and frequency of CD25+ CD127lo Tregs was compared by flow cytometry. Untreated cells were used as control. Each dot represents biological replicate from an independent human donor. n = 8 for oxLDL and n = 9 for IFN γ. Representative flow cytometry plots are shown on the right. (B) MFI of Annexin V in Tregs from oxLDL or IFN γ treated PBMC's. Untreated cells were used as control. MFI of Annexin V in untreated cells was normalized to 100 and relative MFI of treated samples was plotted against it. n = 4. Each dot represents biological replicate from an independent human donor. Representative flow cytometry plots are shown on the right. (C) Representative flow cytometry plots for TMRM staining in Tregs from PBMC's treated with oxLDL, IFN γ or oxLDL + IFN γ. (D) Heatmap comparing the expression of different ER stress and apoptosis related genes in Tregs circulating in blood or residing in plaque of CAD patients (GSE196943). Blood and plaque cells are matched by the same patients. (E) MFI of ER tracker in CD4 + CD8- T cells from healthy and CAD patients. n = 6. Each dot represents an independent donor. Representative histograms for healthy (blue) and CAD (red) are shown on the right. Horizontal bars depict the median. Statistical comparisons were done using paired two-tailed T test in (A, B) and unpaired two-tailed T test in (E). Results are represented as mean ± SEM. Numerical P values are listed at the top of each bar graph. Source data are available online for this figure.

