## [Peer Review File · EMBO Molecular Medicine]

ER stress induced mitochondrial dysfunction drives Treg instability in coronary artery disease

Smriti Parashar, Mohammad Oliaeimotlagh, Payel Roy, Qingkang Lyu, Anusha Bellapu, Mikhail Fomin, Sunil Kumar, Yan Wang, Chantel McSkimming, Coleen McNamara, and Klaus Ley

Corresponding author(s): Klaus Ley (kley@augusta.edu) , Smriti Parashar (sparashar@augusta.edu)

Review Timeline:

Submission Date:	7th Apr 25
Editorial Decision:	13th May 25
Revision Received:	28th Aug 25
Editorial Decision:	22nd Sep 25
Revision Received:	23rd Sep 25
Accepted:	1st Oct 25

Editor: Lise Roth

Transaction Report:

13th May 2025

Dear Dr. Ley,

Thank you for submitting your manuscript to EMBO Molecular Medicine. Please accept my apologies for the delay in responding; one of the reviewers required additional time to complete their report. We have now received feedback from the three reviewers who agreed to evaluate your manuscript. As you will see from the reports below, they acknowledge the interest of the study and support publication of your work, provided that appropriate major revisions are made.

In order for us to consider the manuscript further, it is necessary that you address the reviewers' concerns in full. Following further discussion within the team and with the referees, we agreed that the revisions should focus on functionally validating the proposed mechanism. As obtaining additional human samples may be challenging, this will NOT be requested for further consideration; however, limitations should be discussed in the manuscript.

EMBO Molecular Medicine only allows a single round of revisions, and acceptance or rejection of the manuscript will depend on how complete your responses are in the final version. For this reason, and to save you frustration later on, I would strongly discourage you from submitting an incomplete revision.

We are expecting your revised manuscript within three to four months, if you anticipate any delay, please contact us.

We require:

- 1) A .docx formatted version of the manuscript text (including legends for main figures, EV figures and tables). Please make sure that the changes are highlighted to be clearly visible.
- 2) Individual production quality figure files as .eps, .tif, .jpg (one file per figure). For guidance, download the 'Figure Guide PDF' (<https://www.embopress.org/page/journal/17574684/authorguide#figureformat>).
- 3) At EMBO Press we ask authors to provide source data for the main figures. Our source data coordinator will contact you to discuss which figure panels we would need source data for and will also provide you with helpful tips on how to upload and organize the files.
- 4) A .docx formatted letter INCLUDING the reviewers' reports and your detailed point-by-point responses to their comments. As part of the EMBO Press transparent editorial process, the point-by-point response is part of the Review Process File (RPF), which will be published alongside your paper.
- 5) A complete author checklist, which you can download from our author guidelines (<https://www.embopress.org/page/journal/17574684/authorguide#submissionofrevisions>). Please insert information in the checklist that is also reflected in the manuscript. The completed author checklist will also be part of the RPF.
- 6) All Materials and Methods need to be described in the main text using our 'Structured Methods' format. According to this format, the Methods section includes a Reagents and Tools Table (listing key reagents, experimental models, software and relevant equipment and including their sources and relevant identifiers) followed by a Methods and Protocols section describing the methods, ideally using a step-by-step protocol format. The aim is to facilitate adoption of the methodologies across labs. Please download and fill our Reagents and Tools Table template (.docx), which you can find in our author guidelines: <https://www.embopress.org/page/journal/14693178/authorguide#structuredmethods>. When submitting your revised manuscript, please do not include the Reagents and Tools Table in the Methods section of the manuscript but upload it as a separate file choosing the file type "Reagent Table".
- 7) Please note that all corresponding authors are required to supply an ORCID ID for their name upon submission of a revised manuscript.
- 8) It is mandatory to include a 'Data Availability' section after the Materials and Methods. Before submitting your revision, primary datasets produced in this study need to be deposited in an appropriate public database, and the accession numbers and database listed under 'Data Availability'. Please remember to provide a reviewer password if the datasets are not yet public (see

<https://www.embopress.org/page/journal/17574684/authorguide#dataavailability>).

9) For data quantification: please specify the name of the statistical test used to generate error bars and P values, the number (n) of independent experiments (specify technical or biological replicates) underlying each data point and the test used to calculate p-values in each figure legend. The figure legends should contain a basic description of n, P and the test applied. Graphs must include a description of the bars and the error bars (s.d., s.e.m.). Please provide exact p values.

10) Our journal encourages inclusion of *data citations in the reference list* to directly cite datasets that were re-used and obtained from public databases. Data citations in the article text are distinct from normal bibliographical citations and should directly link to the database records from which the data can be accessed. In the main text, data citations are formatted as follows: "Data ref: Smith et al, 2001" or "Data ref: NCBI Sequence Read Archive PRJNA342805, 2017". In the Reference list, data citations must be labeled with "[DATASET]". A data reference must provide the database name, accession number/identifiers and a resolvable link to the landing page from which the data can be accessed at the end of the reference. Further instructions are available at .

11) We replaced Supplementary Information with Expanded View (EV) Figures and Tables that are collapsible/expandable online. EV Figures should be cited as 'Figure EV1, Figure EV2' etc... in the text and their respective legends should be included in the main text after the legends of regular figures.

12) The paper explained: EMBO Molecular Medicine articles are accompanied by a summary of the articles to emphasize the major findings in the paper and their medical implications for the non-specialist reader. Please provide a draft summary of your article highlighting

13) Author contributions: CRediT has replaced the traditional author contributions section because it offers a systematic machine readable author contributions format that allows for more effective research assessment. Please remove the Authors Contributions from the manuscript and use the free text boxes beneath each contributing author's name in our system to add specific details on the author's contribution. More information is available in our guide to authors.

Please also suggest a visual abstract to illustrate your article as a PNG file 550 px wide x 300-600 px high. A cropped portion of this image will serve as thumbnail for the table of content on our webpage.

16) As part of the EMBO Publications transparent editorial process initiative (see our Editorial at <http://embomolmed.embopress.org/content/2/9/329>), EMBO Molecular Medicine will publish online a Review Process File (RPF) to accompany accepted manuscripts.

In the event of acceptance, this file will be published in conjunction with your paper and will include the anonymous referee reports, your point-by-point response and all pertinent correspondence relating to the manuscript. Let us know whether you agree with the publication of the RPF and as here, if you want to remove or not any figures from it prior to publication.

I look forward to receiving your revised manuscript.

Yours sincerely,

Lise Roth

***** Reviewer's comments *****

Referee #1 (Comments on Novelty/Model System for Author):

This manuscript presents an intriguing model linking ER stress to mitochondrial dysfunction and Treg instability in CAD. The human relevance and mechanistic novelty are compelling. However, direct evidence substantiating the proposed sequence of ER stress → mitophagy failure → FOXP3 loss remains limited. With additional experimental validation, this study could significantly advance our understanding of immune dysregulation in atherosclerosis.

Referee #1 (Remarks for Author):

This manuscript addresses an understudied mechanism of Treg plasticity in the context of coronary artery disease (CAD). Specifically, it proposes that unresolved endoplasmic reticulum (ER) stress drives mitochondrial dysfunction, impairs mitophagy, and triggers integrated stress responses (ISR) that destabilize FOXP3 expression and promote the emergence of cytotoxic exTregs. The study combines human clinical samples, in vitro modeling, and transcriptional profiling to suggest that ER stress underlies Treg instability in atherosclerotic inflammation. The topic is both timely and potentially impactful, as it may illuminate novel immunometabolic targets for restoring immune tolerance in CAD. Despite the conceptual innovation and translational relevance, several aspects of the mechanistic link between ER stress and mitochondrial dysfunction in Tregs remain insufficiently substantiated. Additional functional validation is necessary to support the proposed causal relationships.

Major Comments:

1. Mechanistic Link Between ER Stress and Mitochondrial Dysfunction

The central hypothesis-ER stress drives mitochondrial dysfunction in exTregs-is insufficiently supported. Data showing concurrent ER stress and mitochondrial changes in exTregs are correlative. The authors should experimentally validate causality. For instance, treatment of Tregs with tunicamycin followed by analysis of mitochondrial parameters (TMRM, MitoSOX, and PINK1 accumulation) could directly link ER stress to mitochondrial collapse.

Additionally, the effect of dual exposure to oxLDL and IFN γ on ER stress markers (e.g., BiP, CHOP, spliced XBP1) should be assessed to strengthen the claim that CAD-like conditions induce ER stress in Tregs. This could be complemented by quantitative PCR or protein-level assessments of canonical ER stress and ISR effectors (ATF4, HSPA5, CHOP/DDIT3).

2. Mitophagy Assessment

The conclusion that exTregs exhibit impaired mitophagy is primarily based on downregulation of mitophagy-related transcripts. Since mitophagy is largely controlled through post-translational mechanisms, functional assessments are necessary. For instance:

- Do exTregs show accumulation of dysfunctional mitochondria?
- Is PINK1 localized to depolarized mitochondria?
- Are these mitochondria successfully engulfed by autophagosomes?

Furthermore, morphological characterization (fragmented/swollen mitochondria) and mass quantification should be included to

reinforce mitochondrial dysfunction in exTregs.

3. Source and Identity of exTregs

There is ambiguity about the physiological relevance of the exTregs analyzed in early figures. Many analyses appear to use exTregs derived from healthy PBMCs, where such populations are rare. The study would benefit from directly isolating Tregs and exTregs from CAD patient blood and comparing ER stress, mitochondrial, and functional parameters. This would validate whether observed phenotypes reflect in vivo pathogenic exTregs.

4. Functional Consequences of Treg Destabilization

The downstream implications of ER stress-induced instability are underexplored. The authors could:

- Assess whether FOXP3 loss is tied to CNS2 methylation status in tTregs vs iTregs under ER stress.
 - Compare ER stress responses between iTregs and tTregs, ideally testing whether tunicamycin destabilizes FOXP3 in both subsets.
 - Explore whether ISR-mediated metabolic rewiring (e.g., glycolytic vs oxidative programs) contributes to functional loss.
- Currently, claims about metabolic remodeling are not supported by direct quantification of TCA or glycolytic enzyme expression.

5. Quantification Consistency and Experimental Clarity

- TMRM quantification is inconsistent across Figures 2A, 2C, 3G, and 3K-some relative to MTG, others standalone. Justification and standardization are needed.
- Clarify normalization baselines for all "relative MFI" and "relative frequency" data.
- Indicate which genes are highlighted in volcano plots (e.g., Figure 1A), and ensure legends match main figure formatting.
- Specify sample overlap in data displayed in Figures 4H-4I and analyze potential correlations between ER expansion and mitochondrial depolarization.

Minor Revisions:

- Clearly explain CD57 as a marker-its significance in senescence or cytotoxicity should be contextualized.
- The concentration and duration of tunicamycin and mitochondrial decoupler treatments seem harsh; discuss cell viability post-treatment.
- Revise Figure legends for expanded view figures to include fold-change color scales as in Figure 4K.
- Page 7, line 16: Reference to complex III activity is unsupported without ETC quantification.
- Page 9, lines 1-5: Metabolic instability is discussed but not quantified.
- Page 9, line 6: Use consistent terminology-CVD or CAD.
- Page 10, line 2: Correct DDIT4 → DDIT3 for CHOP encoding.

Referee #2 (Comments on Novelty/Model System for Author):

Technical quality of the experiments, based on the figures and the methods section, seems to be adequate. The novelty is also high because there are very few reports on the biology of exTregs in humans.

Referee #2 (Remarks for Author):

In this work, Parashar S. et al. explore the involvement of endoplasmic reticulum (ER) stress on the de-differentiation of human Treg cells into exTregs and its relationship to mitochondrial dysfunction. This is an interesting manuscript that demonstrates the involvement of these pathways in the acquisition of the exTreg phenotype. Overall the manuscript is well written, and the data exciting. Please see a few comments below.

- Figure 1F. Please show a representative histogram of the expression of CHOP in Tregs vs exTregs, with a positive control for ER stress (e.g. tunicamycin).
- Representative flow stainings should be included in Figures 3 and 4 for all markers examined, not just bar graphs.
- The clinical relevance of the data presented is weak, mostly due to the very small number of participants in each of the two groups (healthy, CAD). In fact, the frequency of Tregs between healthy and CAD patients is not statistically significant, but there is a trend towards increased in frequency in CAD patients. Please increase the number of donors per group.
- Figure 4J. Please group the two panels into one and analyze the data together (two-way ANOVA with correction for multiple comparisons) to have a better view of the data, once you have increased the group size.
- In order for Figure 1K to be more relevant, the authors should also show whether ER stress is seen on the CD4+ T cells from CAD patients compared to healthy individuals in their cohort.

Referee #3 (Comments on Novelty/Model System for Author):

To investigate the role of ER stress as a key inducer of Treg activation, the authors utilized blood samples from CAD patients. While the model system and technical quality of the manuscript are appropriate, I have reservations about its novelty and potential medical impact. Since the role of ER stress in cardiovascular diseases is well studied and published.

Referee #3 (Remarks for Author):

Parashar et al., in their manuscript titled "Endoplasmic reticulum stress-induced mitochondrial dysfunction drives Treg instability in patients with coronary artery disease," investigate the role of ER stress in promoting instability of regulatory T cells (Tregs) in CAD patients. The authors propose that the inflammatory environment in atherosclerosis triggers ER stress and mitochondrial dysfunction, leading to Treg instability. While the study presents a potentially interesting concept, several aspects lack clarity and would benefit from further elaboration.

Major Comments:

1. Figure 1 clearly demonstrates increased ER stress in Tregs. However, the comparison in Figure 1A is somewhat unclear. Are the genes shown as upregulated and downregulated in ex-Tregs relative to Tregs? Please clarify the comparative basis and label the expression changes accordingly.
2. Although the authors discuss the role of protein folding in ER stress, no direct evidence is provided. Inclusion of western blotting or gene expression data for key unfolded protein response (UPR) markers would significantly strengthen the manuscript.
3. In Figure 2, the authors suggest that mitochondrial dysfunction is triggered by ER stress, but the only supporting data presented is TMRM staining. It is intriguing that tunicamycin increases TMRM intensity at 24 hours but decreases it at 48 hours, implying that prolonged ER stress may be harmful. It would be valuable to visualize the mitochondrial network morphology at both time points.
4. What type of mitochondrial dysfunction results from prolonged ER stress? Is there a reduction in mitochondrial number, or merely a decline in function? It would be informative to assess mitochondrial content and compare this between 24 and 48 hours, along with expression or activity of mitochondrial complexes, such as Complex I and IV.
5. The manuscript lacks discussion of the mitochondrial stress response, particularly involving ATF4 and ATF5. Please evaluate and incorporate data on these transcription factors, as they are relevant to both ER and mitochondrial stress signaling.
6. Figure 2E indicates a downregulation of mitophagy-related transcripts, suggesting impaired mitochondrial clearance. This should be compared to the total mitochondrial load in cells. Furthermore, p62 accumulation is typically observed in blocked autophagy—since mitophagy markers are decreased, can the authors include protein-level analyses (e.g., western blot or immunostaining) for p62 and other autophagy markers?
7. Figure 3 shows that prolonged ER and mitochondrial stress leads to upregulation of apoptosis markers. What is the status of ER-mitochondria contact sites (MAMs) under these conditions? It would be important to assess whether protein folding and transport disruptions at these junctions contribute to the observed apoptotic response.

***** Reviewer's comments *****

Referee #1 (Comments on Novelty/Model System for Author):

This manuscript presents an intriguing model linking ER stress to mitochondrial dysfunction and Treg instability in CAD. The human relevance and mechanistic novelty are compelling. However, direct evidence substantiating the proposed sequence of ER stress → mitophagy failure → FOXP3 loss remains limited. With additional experimental validation, this study could significantly advance our understanding of immune dysregulation in atherosclerosis.

Referee #1 (Remarks for Author):

This manuscript addresses an understudied mechanism of Treg plasticity in the context of coronary artery disease (CAD). Specifically, it proposes that unresolved endoplasmic reticulum (ER) stress drives mitochondrial dysfunction, impairs mitophagy, and triggers integrated stress responses (ISR) that destabilize FOXP3 expression and promote the emergence of cytotoxic exTregs. The study combines human clinical samples, in vitro modeling, and transcriptional profiling to suggest that ER stress underlies Treg instability in atherosclerotic inflammation. The topic is both timely and potentially impactful, as it may illuminate novel immunometabolic targets for restoring immune tolerance in CAD. Despite the conceptual innovation and translational relevance, several aspects of the mechanistic link between ER stress and mitochondrial dysfunction in Tregs remain insufficiently substantiated. Additional functional validation is necessary to support the proposed causal relationships.

We thank the reviewer for aptly summarizing the relevance of our findings and for their constructive feedback. Based on the reviewer's suggestions, we have now incorporated new experiments to establish a direct link between ER stress and mitochondrial dysfunction in Tregs, as well as other functional analyses. The detailed response follows.

Major Comments:

1. Mechanistic Link Between ER Stress and Mitochondrial Dysfunction

The central hypothesis-ER stress drives mitochondrial dysfunction in exTregs-is insufficiently supported. Data showing concurrent ER stress and mitochondrial changes in exTregs are correlative. The authors should experimentally validate causality. For instance, treatment of Tregs with tunicamycin followed by analysis of mitochondrial parameters (TMRM, MitoSOX, and PINK1 accumulation) could directly link ER stress to mitochondrial collapse.

Response 1a: TMRM retention in Tunicamycin treated Tregs

To address this, we have incorporated new data where we treated PBMCs from healthy human donors with Tunicamycin and analyzed CD25+CD127^{lo} Tregs for retention of TMRM. We found that prolonged (48h Tunicamycin) ER stress led to a significant loss of TMRM retention in Tregs, suggesting that persistent unresolved ER stress can cause loss of mitochondrial membrane potential in Tregs.

Frequency of TMRM- Tregs

Response 1b: Mitos accumulation in Tunicamycin treated Tregs

We also analyzed the accumulation of mitochondrial reactive oxygen species in these samples by staining with Mitosox and found a significant increase in the frequency of Tregs staining positively for Mitosox in 48h-Tunicamycin treated samples.

Frequency of MitoSOX+ Tregs

Additionally, we observed that almost all MitoSOX+ cells were TMRM- suggesting that the generation of mitochondrial ROS is driving the loss of mitochondrial membrane potential in these cells. This is consistent with a previous study that showed that Mitosox accumulation could reproducibly trigger mitochondrial depolarization in cardiac myocytes (PMID: 11015441).

Response 1c: PINK1 status in Tunicamycin treated Tregs

Previous studies have shown that Tunicamycin can decrease PINK1 expression in lung cells by transcriptional repression via ATF3. This defect in PINK1 expression disrupts mitochondrial homeostasis and promotes lung fibrosis (PMID: 25562319; 29363258) . We analyzed PINK1 expression in Tregs treated with Tunicamycin by western blotting and found reduced expression of PINK1 as early as 24h of Tunicamycin treatment.

We think that reduced expression of PINK1 triggered by Tunicamycin-mediated ER stress further disrupts mitochondrial homeostasis in treated Tregs.

Additionally, the effect of dual exposure to oxLDL and IFN γ on ER stress markers (e.g., BiP, CHOP, spliced XBP1) should be assessed to strengthen the claim that CAD-like conditions induce ER stress in Tregs. This could be complemented by quantitative PCR or protein-level assessments of canonical ER stress and ISR effectors (ATF4, HSPA5, CHOP/DDIT3).

Response: To strengthen our claim that CAD-like conditions can induce ER stress and mitochondrial dysfunction in Tregs, we analyzed the expression of BiP, spliced XBP1 and ATF4 in Tregs treated with OxLDL, IFN γ or OxLDL+IFN γ by western blotting. Indeed, the most prominent increase in these markers was observed on dual exposure. The fold change expression in each of these markers normalized to Actin is indicated at the bottom of each blot. Additionally, all these treatments led to enhanced frequency of Tregs with low TMRM retention, the most consistent effect being observed with dual exposure.

2. Mitophagy Assessment

The conclusion that exTregs exhibit impaired mitophagy is primarily based on downregulation of mitophagy-related transcripts. Since mitophagy is largely controlled through post-translational mechanisms, functional assessments are necessary. For instance:

- Do exTregs show accumulation of dysfunctional mitochondria?
- Is PINK1 localized to depolarized mitochondria?
- Are these mitochondria successfully engulfed by autophagosomes?

Response 2a: Accumulation of dysfunctional mitochondria in exTregs

A significantly higher percentage of exTregs showed accumulation of dysfunctional mitochondria (analyzed by low TMRM retention) compared to the respective Tregs from same donors.

Response 2b and 2c: Is PINK1 localized to depolarized mitochondria? Are these mitochondria successfully engulfed by autophagosomes?

The very low frequency of exTregs in human PBMCs (0.1-0.2% of total CD3+CD4+CD8- T cells) precludes us from doing a detailed confident microscopic analysis of these cells. Concurrent analysis of depolarized mitochondria (TMRM negative) and Pink1/LC3B cannot be done as TMRM staining is incompatible with fixation-permeabilization that is a requisite

for intracellular staining of Pink1/LC3b. We have now mentioned it as the limitation of this study.

However, to confirm that exTregs are defective in mitophagy, we compared accumulation of p62 (a hallmark of defective mitophagy/autophagy) in Tregs vs exTregs and found enhanced p62 protein accumulation in exTregs. This finding along with accumulation of dysfunctional mitochondria in exTregs supports our conclusion that exTregs are defective in mitophagy.

Furthermore, morphological characterization (fragmented/swollen mitochondria) and mass quantification should be included to reinforce mitochondrial dysfunction in exTregs.

exTregs showed enhanced total mitochondrial mass as revealed by significantly higher mitotracker staining compared to Tregs from the same donors.

While very low frequency of exTregs in human PBMC's limits us from doing a confident quantitative analysis of these cells by imaging, mitochondrial morphology in exTregs appeared to be swollen compared to Tregs as revealed by qualitative analysis by live cell confocal imaging.

We also compared the metabolic profile of Tregs vs exTregs by SCENITH (a flow cytometry-based assay for metabolic studies of rare cells ex vivo (PMID: 33264598)). Briefly, this assay was developed on the hypothesis that almost half of the cellular energy production is utilized for protein synthesis. Therefore, this assay monitors rapid changes in protein translation (determined by fluorescently tagged Puromycin incorporation) upon inhibition of a metabolic pathway.

Consistent with the loss of mitochondrial function in exTregs, they showed reduced mitochondrial dependence as revealed by lower sensitivity of protein translation to Oligomycin-mediated Electron transport chain inhibition in exTregs compared to Tregs.

Taken together we believe that exTregs have mitochondrial dysfunction based on the following: 1) Loss of mitochondrial membrane potential as revealed by lower TMRM retention 2) Higher mitochondrial ROS, 3) higher mitochondrial mass possibly as a compensatory effect and/or due to impaired autophagic clearance 4) reduced mitochondrial dependence in exTregs compared to Tregs as revealed by SCENITH profiling.

3. Source and Identity of exTregs

There is ambiguity about the physiological relevance of the exTregs analyzed in early figures. Many analyses appear to use exTregs derived from healthy PBMCs, where such populations are rare. The study would benefit from directly isolating Tregs and exTregs from CAD patient blood and comparing ER stress, mitochondrial, and functional parameters. This would validate whether observed phenotypes reflect in vivo pathogenic exTregs.

We compared Tregs vs exTregs in PBMCs from CAD patients and found no significant decrease in TMRM/MTG ratio in patient exTregs compared to patient Tregs, suggesting that in CAD patients, some of the changes characteristic of exTregs are already happening in Tregs.

We think that nutritional and inflammatory stressors in atherosclerosis can drive Tregs to gain more exTreg-like phenotype. Indeed, studies have shown that Tregs from CAD patients are less suppressive and more inflammatory (PMID: 26260103, Burkard et al, European heart Journal, Vol.44, suppl. 2, 2023), a phenotype that we have previously described as a hallmark of exTregs (PMID: 37563308).

4. Functional Consequences of Treg Destabilization

The downstream implications of ER stress-induced instability are underexplored. The authors could:

- Assess whether FOXP3 loss is tied to CNS2 methylation status in tTregs vs iTregs under ER stress.
- Compare ER stress responses between iTregs and tTregs, ideally testing whether tunicamycin destabilizes FOXP3 in both subsets.

We isolated nTregs (the majority of which are thymus-derived, Sakaguchi, PMID: 32367041) and generated in vitro-induced Tregs (iTregs) from the same donors and compared FOXP3 expression in both subsets after Tunicamycin treatment. A comparable decrease in FOXP3 expression was observed in both subsets.

It would be interesting to determine if this loss is tied to CNS2 methylation. Tunicamycin is a potent inducer of ER stress and upregulates ATF4 (PMID: 33646118). ATF4 can negatively regulate NRF1-TFAM signaling and promote mitochondrial dysfunction (PMID: 33177163). TFAM deficiency has previously been shown to cause enhanced methylation of TSDR of the *Foxp3* locus (PMID: 31269437), therefore promoting Treg instability. Taken together in context, it is tempting to speculate that ER stress may negatively regulate FOXP3 expression in Tregs by hypermethylation of *Foxp3* TSDR. However, a detailed exploration of this connection is beyond the scope of current manuscript and is now discussed as a future perspective.

- Explore whether ISR-mediated metabolic rewiring (e.g., glycolytic vs oxidative programs) contributes to functional loss. Currently, claims about metabolic remodeling are not supported by direct quantification of TCA or glycolytic enzyme expression.

To directly address if exTregs have undergone metabolic rewiring, we compared the mitochondrial dependence of Tregs vs exTregs by SCENITH (a flow cytometry based assay for metabolic studies of rare cells ex vivo, PMID: 33264598).

Consistent with the loss of mitochondrial function in exTregs, they showed reduced mitochondrial dependence as revealed by lower sensitivity of protein translation to Oligomycin-mediated ETC inhibition compared to Tregs.

We also observed enhanced protein expression of ATF4 in exTregs. ATF4 can induce metabolic reprogramming by regulating the expression of PCK2, an enzyme that can convert TCA intermediates to glycolytic intermediates (PMID: 28566324), suggesting that reduced mitochondrial dependence in exTregs may be due to ISR-mediated metabolic rewiring.

5. Quantification Consistency and Experimental Clarity

- **TMRM quantification is inconsistent across Figures 2A, 2C, 3G, and 3K-some relative to MTG, others standalone. Justification and standardization are needed.**

Mitochondrial analysis in this study with most donors was initially done as a standalone TMRM analysis to determine loss of mitochondrial membrane potential. However, later, we came across references (eg. PMID: 38260521) that suggested normalization to MTG to account for compensation in TMRM signal due to enhanced number of mitochondria, if any).

Apart from Fig. 3j, (Fig. 3g in previous version) where we did not have MTG in the staining panel as it was done during initial phase of the study, all the data in current manuscript is represented as TMRM/MTG wherever the MFI of TMRM is plotted.

Data in previous Fig. 2c (which is now Fig. 2d) was repeated with TMRM/MTG for 3 donors and was shown in Expanded view Figure 2b (now EV figure 2h) in previously submitted manuscript. Previous figures 2a and 3k were represented as TMRM/MTG in the previously submitted version.

- **Clarify normalization baselines for all "relative MFI" and "relative frequency" data.**

Average of the control data points from an experiment was set to 100 and then all control and test data was plotted against it. This has now been clarified in figure legends.

- **Indicate which genes are highlighted in volcano plots (e.g., Figure 1A), and ensure legends match main figure formatting.**

In Fig. 1A, ER stress related genes are highlighted in asterisks and signature genes for exTregs and Tregs are boxed. This is now mentioned in the figure legends.

- **Specify sample overlap in data displayed in Figures 4H-4I and analyze potential correlations between ER expansion and mitochondrial depolarization.**

These data are from the same donors suggesting a correlation between ER expansion and mitochondrial depolarization.

Minor Revisions:

- Clearly explain CD57 as a marker-its significance in senescence or cytotoxicity should be contextualized.

CD57 is considered a marker for replicative senescence in CD4+ and CD8+ T cells with such cells showing reduced proliferative potential, high secretion of TNF α and IFN γ and enhanced expression of Perforin and Granzymes (PMID: 12433688, PMID: 16339584,

PMID: 17015699, PMID: 18820174, PMID: 26850637). Indeed, our previous work has shown human exTregs to exhibit all these traits (PMID: 37563308).

We suggest that the gain of CD57 expression and resultant loss of proliferative potential in exTregs further exacerbates cellular stress in these cells. This is because cell division is known to contribute to the dissolution of proteotoxic stress by clearing protein aggregates or by assymmetrically distributing them between daughter cells (PMID: 40476343, PMID: 25620549, PMID: 25417105). Inability of exTregs to divide further worsens their ability to resolve stress.

- The concentration and duration of tunicamycin and mitochondrial decoupler treatments seem harsh; discuss cell viability post-treatment.

For Tunicamycin, 5-10% of CD3+ T cells die after treatment. For TBHP, 20% of CD3+ T cells die post- treatment

- Revise Figure legends for expanded view figures to include fold-change color scales as in Figure 4K.

We tried to do that, but heat maps generated by Gene set enrichment analysis (GSEA) do not indicate numerical fold-change for color scales. The program explains their color scale as follows: “In a heat map, expression values are represented as colors, where the range of colors (red, pink, light blue, dark blue) shows the range of expression values (high, moderate, low, lowest)”. The color scale in each row represents the GSEA-algorithm defined row minimum (blue) to row maximum (red) for expression level of individual gene in every row.

Therefore, we have now mentioned the statement, “Color scale in the heat map is based on GSEA row minimum (blue) to row maximum (red) in the legends.

- Page 7, line 16: Reference to complex III activity is unsupported without ETC quantification.

This reference has now been removed.

- Page 9, lines 1-5: Metabolic instability is discussed but not quantified.

We have now quantified metabolic instability in exTregs by SCENITH

- Page 9, line 6: Use consistent terminology-CVD or CAD.

We now consistently use CAD.

- Page 10, line 2: Correct DDIT4 → DDIT3 for CHOP encoding.

Thank you for pointing this, this has been corrected.

Referee #2 (Comments on Novelty/Model System for Author):

Technical quality of the experiments, based on the figures and the methods section, seems to be adequate. The novelty is also high because there are very few reports on the biology of exTregs in humans.

Referee #2 (Remarks for Author):

In this work, Parashar S. et al. explore the involvement of endoplasmic reticulum (ER) stress on the de-differentiation of human Treg cells into exTregs and its relationship to mitochondrial dysfunction. This is an interesting manuscript that demonstrates the involvement of these pathways in the acquisition of the exTreg phenotype. Overall, the manuscript is well written, and the data exciting. Please see a few comments below.

We thank the reviewer for highlighting the novelty of this study and for their helpful comments. We have now included new data to address these suggestions. The detailed response follows.

- Figure 1F. Please show a representative histogram of the expression of CHOP in Tregs vs exTregs, with a positive control for ER stress (e.g. tunicamycin).

This has now been included.

- Representative flow stainings should be included in Figures 3 and 4 for all markers examined, not just bar graphs.

We have included representative flow plots for all data shown in the revised manuscript.

- The clinical relevance of the data presented is weak, mostly due to the very small number of participants in each of the two groups (healthy, CAD). In fact, the frequency of Tregs between healthy and CAD patients is not statistically significant, but there is a trend towards increased in frequency in CAD patients. Please increase the number of donors per group.

We increased the number of donors in each group and still observe mitochondrial dysfunction and ER stress in patient Tregs compared to healthy Tregs. Specifically, the ER tracker signal is significantly higher in CAD patients compared to controls, and the TMRM/MTG ratio is significantly lower.

Additionally, after increasing the number of donors per group, we observe a significant increase in Treg numbers in PBMCs of CAD patients compared to healthy donors. This is consistent with a previous report that shows enhanced Treg numbers in aged CAD patients compared to the healthy controls (Burkard et al, European heart Journal, Vol.44, suppl. 2, 2023). We think that this may be a compensatory mechanism to overcome the functional defects in patient Tregs.

- Figure 4J. Please group the two panels into one and analyze the data together (two-way ANOVA with correction for multiple comparisons) to have a better view of the data, once you have increased the group size.

We have now analyzed the data as suggested and can conclude that Tregs from CAD patients show mitochondrial dysfunction compared to healthy Tregs. Additionally, mitochondrial dysfunction in CAD Tregs is comparable to CAD exTregs suggesting that patient Tregs may have started to gain an exTreg-like phenotype.

- In order for Figure 1K to be more relevant, the authors should also show whether ER stress is seen on the CD4+ T cells from CAD patients compared to healthy individuals in their cohort.

We compared the ER tracker staining in CD4+ T cells among healthy vs CAD patients and found significantly enhanced staining in CD4+T cells from CAD patients.

Referee #3 (Comments on Novelty/Model System for Author):

To investigate the role of ER stress as a key inducer of Treg activation, the authors utilized blood samples from CAD patients. While the model system and technical quality of the manuscript are appropriate, I have reservations about its novelty and potential medical impact. Since the role of ER stress in cardiovascular diseases is well studied and published.

Referee #3 (Remarks for Author):

Parashar et al., in their manuscript titled "Endoplasmic reticulum stress-induced mitochondrial dysfunction drives Treg instability in patients with coronary artery disease," investigate the role of ER stress in promoting instability of regulatory T cells (Tregs) in CAD patients. The authors propose that the inflammatory environment in atherosclerosis triggers ER stress and mitochondrial dysfunction, leading to Treg instability. While the study presents a potentially interesting concept, several aspects lack clarity and would benefit from further elaboration.

We thank the reviewer for nicely highlighting our findings and for their key suggestions. Based on them, we have now included new experiments to further support our findings. The detailed response follows.

Additionally, while we agree that role of ER stress in CAD is well studied, we would like to emphasize that these studies focus on cardiomyocytes, endothelial cells and macrophages. To the best of our knowledge, the role of ER stress in driving Treg instability in CAD has not been studied. Since Treg dysfunction is increasingly being reported in CAD patients (PMID: 26260103, Burkard et al, European heart Journal, Vol.44, suppl. 2, 2023),

our study highlights the role of ER stress and mitochondrial dysfunction as contributing factors towards disturbed Treg homeostasis in CAD.

Major Comments:

1. Figure 1 clearly demonstrates increased ER stress in Tregs. However, the comparison in Figure 1A is somewhat unclear. Are the genes shown as upregulated and downregulated in ex-Tregs relative to Tregs? Please clarify the comparative basis and label the expression changes accordingly.

Yes, the genes are shown as upregulated and downregulated in ex-Tregs relative to Tregs. The labeling in the figure has been changed to improve clarity.

2. Although the authors discuss the role of protein folding in ER stress, no direct evidence is provided. Inclusion of western blotting or gene expression data for key unfolded protein response (UPR) markers would significantly strengthen the manuscript.

To address that protein misfolding is contributing to ER stress in exTregs, we have compared the accumulation of misfolded proteins in Tregs vs exTregs by staining with Proteostat (a dye that specifically intercalates into the cross-beta spine of quaternary protein structures typically found in misfolded and aggregated proteins, which will inhibit the dye's rotation and lead to a strong fluorescence). Indeed, we found enhanced staining

of proteostat in exTregs vs Tregs. MG132, a proteasomal inhibitor was used as a positive control.

Additionally, Gene set enrichment analysis revealed significant enrichment of Hallmark gene signature for unfolded protein response in exTregs compared to Tregs. exTregs also showed enhanced protein expression of Ire1 and PERK by flow cytometry. Please note that very low frequency of exTregs in peripheral human blood precludes us from doing a western blot analysis of these samples. This has now been mentioned as a limitation in the manuscript.

HALLMARK Unfolded protein response

3. In Figure 2, the authors suggest that mitochondrial dysfunction is triggered by ER stress, but the only supporting data presented is TMRM staining. It is intriguing that tunicamycin increases TMRM intensity at 24 hours but decreases it at 48 hours, implying that prolonged ER stress may be harmful. It would be valuable to visualize the mitochondrial network morphology at both times.

To support our conclusion that prolonged ER stress can trigger mitochondrial dysfunction, we compared the frequency of Tregs that actually showed the loss of mitochondrial

membrane potential (TMRM-) and accumulation of Mito ROS (MitoSOX+) after 24 and 48h of Tunicamycin treatment. We did not see a significant change in either of these parameters at 24h, but an approx. 2.5-fold increase in TMRM- and Mitosox+ Tregs at 48h.

Additionally, we observed that almost all Mitosox+ cells were TMRM- suggesting that the generation of mitochondrial ROS is contributing towards the loss of mitochondrial membrane potential in these cells. This is consistent with a previous study that showed that Mitosox accumulation could reproducibly trigger mitochondrial depolarization in cardiac myocytes (PMID: 11015441).

We also analyzed mitochondrial morphology in Tregs treated with Tunicamycin by live cell confocal microscopy and observed a subtle but significant increase in percent of cells showing tubulated/elongated morphology at 48h of Tunicamycin treatment compared to untreated cells. No significant difference was observed between 24 and 48h of Tunicamycin treatment. These observations are consistent with studies showing mitochondrial elongation as a structural adaptation to facilitate mitochondrial respiration under cellular stress (PMID: 29539413, PMID: 29950571).

Please note that during our experimental conditions, approximately 15% of cells are TMRM-MitoSox+ after 48h, suggesting that while ER stress has begun to induce mitochondrial dysfunction, the affect is not absolute yet and the cells still have the capability to adapt to this stress. However, we speculate that this adaptability may not sustain during chronic inflammatory pathologies, wherein Tregs are exposed to ER stress much longer than our experimental conditions.

4. What type of mitochondrial dysfunction results from prolonged ER stress? Is there a reduction in mitochondrial number, or merely a decline in function? It would be informative to assess mitochondrial content and compare this between 24 and 48 hours, along with expression or activity of mitochondrial complexes, such as Complex I and IV.

In addition to loss of mitochondrial membrane potential and accumulation of MitoRos (discussed in response to comment 3 of the reviewer), we also observed a decrease in mitochondrial mass in Tregs after 48h of Tunicamycin treatment.

However, metabolic profiling of these cells by SCENITH (PMID: 33264598) did not show any considerable change in mitochondrial dependence in these cells at 48h of Tunicamycin treatment suggesting that mitochondria are still functional in these cells.

We hypothesize that while under our experimental conditions, ER stress induced by Tunicamycin has started to trigger mitochondrial dysfunction (approximately 15% cells are TMRM- and Mitosox+ along with decrease in mitochondrial mass), cells begin to compensate for these defects possibly by enhancing the functional capacity of remaining mitochondria. This is supported by our observation of significant increase in percent of cells showing elongated mitochondrial morphology (a previously reported structural adaptation of mitochondria to facilitate mitochondrial respiration under cellular stress, PMID: 29950571).

In line with this hypothesis, a previous study in mouse fibroblasts (PMID: 31023583) has shown that under conditions of ER stress, there is an increase in mitochondrial respiration, Complex I and IV activity and respiratory supercomplex assembly to fulfill energy demands needed to facilitate protein folding under these conditions.

5. The manuscript lacks discussion of the mitochondrial stress response, particularly involving ATF4 and ATF5. Please evaluate and incorporate data on these transcription factors, as they are relevant to both ER and mitochondrial stress signaling.

Thank you for this suggestion. We looked at the expression of ATF4, a transcription factor involved in regulating UPR^{ER} (PMID: 32457508), mitochondrial stress response (PMID: 28566324) as well as integrated stress response (PMID: 27629041) and found enhanced ATF4 protein expression in exTregs.

Additionally, expression of ATF5, a transcription factor previously reported to induce protective mitochondrial UPR (UPR^{mt}) during conditions of mitochondrial stress (PMID: 22700657, PMID: 27426517) was also upregulated in exTregs.

These findings further corroborate our observations that exTregs are undergoing ER and mitochondrial stress.

6. Figure 2E indicates a downregulation of mitophagy-related transcripts, suggesting impaired mitochondrial clearance. This should be compared to the total mitochondrial load in cells. Furthermore, p62 accumulation is typically observed in blocked autophagy—since mitophagy markers are decreased, can the authors include protein-level analyses (e.g., western blot or immunostaining) for p62 and other autophagy markers?

exTregs showed a significant increase in mitochondrial load compared to Tregs as revealed by enhanced mitotracker staining. Additionally, we also saw an enhanced accumulation of p62 protein in exTregs. Taken together, these findings suggest that mitochondrial clearance is impaired in exTregs.

7. Figure 3 shows that prolonged ER and mitochondrial stress leads to upregulation of apoptosis markers. What is the status of ER-mitochondria contact sites (MAMs) under these conditions? It would be important to assess whether protein folding and transport disruptions at these junctions contribute to the observed apoptotic response.

Gene set enrichment analysis revealed enrichment of gene signature of ER-mitochondria contact sites in exTregs. This is consistent with previous reports that ER stress can induce enhanced coupling of ER and mitochondria (PMID: 21628424, PMID: 36158213).

We also analyzed the expression of genes involved in mediating transfer of mitochondrial precursor proteins from ER to mitochondria via ER-SURF pathway (PMID: 38565738; PMID: 30213914). While there are no direct ER-Mitochondria encounter structure (ERMES) homologs in humans, the expression of GRAMD1A (mammalian homolog of Lam6, that is required to mediate ER-mitochondria protein transfer) was significantly decreased in exTregs. No change in TOMM70 expression was observed in exTregs.

Additionally, expression of ATP13A1, the P5A-ATPase required for translocation of ER-stranded mitochondrial precursors to mitochondria in humans (PMID: 36283413) was also significantly reduced in exTregs. Expression of TOMM20, another protein reported to mediate ER-mitochondria protein transport (PMID: 33589622) was also downregulated in exTregs. Taken in context with the existing literature, our findings suggest that transport disruptions at ER-mitochondria junctions might be contributing to the pro-apoptotic response in exTregs.

GOCC_Mitochondria_associated_ER_membrane_contact_site

22nd Sep 2025

Dear Dr. Ley,

Thank you for submitting your revised study. We have now received the reports from the referees. As you will see below, they are satisfied with the revisions, and I will therefore be able to accept your manuscript once the following editorial concerns are addressed:

1/ Manuscript text:

- Please accept previous changes and only keep in track changes mode any new modification in the text.
- Please provide up to 5 keywords.
- Methods and Protocols should be renamed Methods.
- Data availability section: Note that the Data Availability Section is restricted to new primary data that are part of this study. In case you have no data that requires deposition in a public database, please state so in this section ("This study includes no data deposited in external repositories"). Please move this section to the end of the Methods section.

2/ Figures:

- Please correct the nomenclature to "Figure EV1" etc. in the legends and labelling in the figure files.
- Appendix: please add page numbers in the table of contents, add "Appendix Figure S1" to the legend, and remove the panel A, which is not needed since there is only one panel. Alternatively, this figure could be a new EV figure, or merged to an existing EV figure.
- Please address the query from our data editors in the figure legends: information related to n is missing in the legend of figure 1A, please correct.

3/ Thank you for providing Source Data. For Fig. 2I, please provide the original unprocessed microscopy images.

4/ In the author checklist, please fill in the section 'Experimental study design and statistics/ inclusion-exclusion criteria'.

5/ Please provide the reagent table in an editable (.docx) format.

6/ Thank you for providing a visual abstract. Please upload it as a tiff/jpeg/PNG file 550 px wide x 300-600 px high and make sure that the text remains legible. A cropped portion of this image will serve as thumbnail for the table of content on our webpage.

Please also provide a synopsis text that should include a short stand first (maximum of 300 characters, including space) as well as 2-5 one-sentences bullet points that summarizes the paper (maximum of 30 words / bullet point).

7/ As part of the EMBO Publications transparent editorial process initiative (see our Editorial at <http://embomolmed.embopress.org/content/2/9/329>), EMBO Molecular Medicine will publish online a Review Process File (RPF) to accompany accepted manuscripts.

This file will be published in conjunction with your paper and will include the anonymous referee reports, your point-by-point response and all pertinent correspondence relating to the manuscript. Let us know whether you agree with the publication of the RPF and as here, if you want to remove or not any figures from it prior to publication.

I look forward to receiving your revised manuscript.

Yours sincerely,

Lise Roth

**** Reviewer's comments ****

Referee #1 (Remarks for Author):

The authors have addressed some of my concerns using basic experiments. Key mechanistic aspects of the paper are still obscure and most data presented are correlative. However, the phenotype is interesting and the study sheds light on the consequences of ER stress in Tregs, which remain largely unexplored.

Referee #2 (Comments on Novelty/Model System for Author):

there are other publications that have shown the role of ER on various aspects of Treg biology

Referee #2 (Remarks for Author):

The authors have adequately addressed all my comments.

Referee #3 (Comments on Novelty/Model System for Author):

The technical quality of the work is high, with appropriate experimental design and statistical analyses supporting the conclusions. The novelty is medium, as the findings build on existing knowledge but add important mechanistic detail. The medical impact is also medium, given the translational relevance but limited immediate clinical applicability. The chosen model system is adequate for the study objectives and faithfully recapitulates key aspects of the disease. Future studies might consider validation in complementary models or patient-derived systems to further strengthen translational impact. No ethical concerns are noted regarding the use of this model organism.

Referee #3 (Remarks for Author):

The authors have addressed all of my questions clearly and satisfactorily. I am pleased with the thorough revisions and believe the manuscript has been significantly strengthened. I congratulate the authors on their work.

All editorial and formatting issues were resolved by the authors.

1st Oct 2025

Dear Dr. Ley,

Thank you for submitting your revised files. I am pleased to inform you that your manuscript is accepted for publication and is now being sent to our publisher to be included in the next available issue of EMBO Molecular Medicine.

Please note that I have edited the first-stand of your synopsis as follows:

"Mitochondrial dysfunction triggered by ER stress was found to make Tregs unstable and to cause them to lose their protective function."

Let us know immediately if you do not agree with these changes.

Yours sincerely,

Lise Roth
